# Disentangling the QiGan Encoded by a DNN towards the Go Game

## Abstract

Given a deep neural network (DNN) that has surpassed human beings in a task, disentangling the explicit knowledge encoded by the DNN to obtain some new insights into the task is a new promising-yet-challenging regime in explainable AI. In this paper, we aim to disentangle the "QiGan[1]" encoded by the AI model for the Go game, which has beat top human players. Specifically, we disentangle primitive shape patterns of stones memorized by the value network, and these shape patterns represent the "QiGan[1]" used to conduct a fast situation assessment of the current board state. The universal-matching property of interactions ensures that human players can learn accurate and verifiable shape patterns, rather than specious intuitive analysis. In experiments, our method explains lots of novel shape patterns beyond traditional shape patterns in human knowledge.

## 1 Introduction

The explanation for AI models has gained increasing attention in recent years. However, in this paper, we consider a new problem, *i.e.*, if an AI model has achieved superior performance to human beings in a task, *how can we clearly disentangle exact inference patterns used by the AI model to help people better understand new hidden rules for the task?* Because AI models for the Go game have been widely regarded to have surpassed human players (Granter et al., 2017; Fang et al., 2018; Intelligence, 2016), in this paper, let us consider the Go game as a case study, and disentangle shape patterns[2] used by AI models[3] to play the Go game.

In fact, the disentanglement of shape patterns is of special values for the Go game. Because the Go game is considered as the most difficult game with the vastest searching space of $10^{171}$ states, which is far beyond the searching ability of human players or computers (Lee, 2004; Van Der Werf, 2004), both people and neural network have to use "QiGan[1]" as a fast intuitive situation assessment. In comparison, other games (*e.g.* chess and poker) primarily rely on search capacities. In the AI model for the Go game, the value network encodes the "QiGan[1]," and the policy network and the Monte Carlo Tree Search correspond to searching capabilities (Gelly & Silver, 2011; Buesing et al., 2020). "QiGan[1]" is also one of the core skills that human players need to learn.

Therefore, we aim to discover explicit new shape patterns[2] of stones (*i.e.*, QiGan[1]) used by the value network to estimate the advantage score without a sophisticated search. The discovered patterns may probably be beyond current human understanding of the Go game. As the mathematical guarantee of the pattern discovery, Li & Zhang (2023) have discovered and Ren et al. (2024a) have proven that the output score of a DNN can usually be decomposed into effects of interactions encoded by a DNN. For example, given a board state $\boldsymbol{x}$ with $n$ stones, $N = \{1, 2, ..., n\}$, the value network in Figure 1 outputs the advantage score $v(\boldsymbol{x})$ of white stones. Each interaction between stones in $S \subseteq N$ represents a shape pattern corresponding to an AND relationship between stones in $S$, which is equivalently encoded by the value network. When all stones in $S$ are present, the interaction $S$ is activated and contributes an effect $I(S)$ to $v(\boldsymbol{x})$. Removing any stone in $S$ will remove the effect $I(S)$ from $v(\boldsymbol{x})$.

---

[1] In the community of the Go game, "QiGan" is widely used to refer to the fast situation assessment based on shape patterns of stones without a sophisticated search. "QiGan" can also be replaced with "knowledge."

[2] Shape patterns, refer to the various shapes formed by the arrangement of stones on the board.

[3] We use the KataGo (Wu, 2019) for testing, because the AlphaGo is not open-sourced.

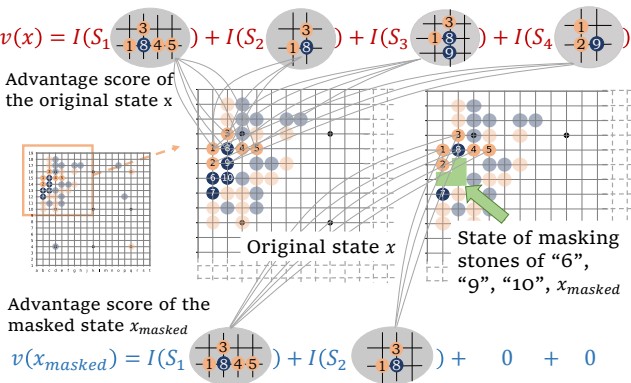

$v(x) = I(S_1 \; ) + I(S_2 \; ) + I(S_3 \; ) + I(S_4 \; )$

Advantage score of
the original state $x$

Original state $x$

State of masking
stones of "6",
"9", "10", $x_{masked}$

Advantage score of the
masked state $x_{masked}$

$v(x_{masked}) = I(S_1 \; ) + I(S_2 \; ) + \quad 0 \quad + \quad 0$

Figure 1: Interactions encoded by the value network. Each interaction $S$ represents a specific shape pattern. Because stones $x_6, x_9, x_{10}$ are removed (masked) from the board, the AND interactions $S_3$ and $S_4$ are deactivated in the masked board state, and interaction effects $I(S_3)$ and $I(S_4)$ are removed from the advantage score $v(x_{masked})$. We can label $T = \{1, 3, 8\}$ as a common coalition that represents a common shape pattern shared by different interactions $S_1, S_2, S_3$.

However, the theory of the previous interaction is no longer effective for the explanation of the Go game due to the superior complexity of the Go game from the following three perspectives.

• First, we prove that the previous interaction cannot well explain the OR relationship between stones encoded by the DNN. The presence of any stones in $S$ activates an OR relationship and contributes an effect $I_{or}(S)$. We prove that the OR pattern would be explained as $2^{|S|}$ different AND interactions.

Therefore, we extend AND interactions to explain OR relationships between stones encoded by the value network. More crucially, **(1) Mathematical guarantee.** We first prove the universal matching property of AND-OR interactions to ensure that OR interactions faithfully explain the value network. *I.e.*, given a board state $x$, the advantage score can be accurately matched by a small number of AND interactions and OR interactions, no matter how we randomly remove stones from the board (see Theorem 2.4, Figure 6). **(2) Experimental verification.** We verify that shape patterns extracted from one board state can also explain the network outputs on other board states.

• Second, we find that the advantage score estimated by the value network is usually shifted, when white stones on the board are far less or far more than black stones on the board. This also hurts the faithfulness of explaining shifted advantage scores. Therefore, we develop a method to alleviate the shifting problem. The faithfulness of the explanation is experimentally verified in Section 2.6, so as to improve the faithfulness/fidelity of the explanation.

• Furthermore, we notice that some shape patterns for the Go game are usually quite complex, *i.e.*, containing a large number of stones. Too complex shape patterns are usually considered as the unusual shape patterns memorized by the value network, instead of a common shape pattern shared by different board states. This boosts the difficulty of understanding the Go game. Thus, we further identify some common combinations of stones that are shared by different interactions/shapes, namely *coalitions*. For example, in Figure 1, the interactions $S_1 = \{1, 3, 4, 5, 8\}, S_2 = \{1, 3, 8\}, S_3 = \{1, 3, 8, 9\}$ all contain the coalition $T = \{1, 3, 8\}$. We apply the attribution method (Xinhao Zheng, 2023) to estimate the attribution of each coalition to help people understand the DNN's logic.

We collaborate with professional human Go players to compare interactions/coalitions encoded by the value network with the human player's QiGan[1] of the Go game, so as to discover shape patterns beyond human understanding. We find that some discovered shape patterns well fit human understanding, but other discovered patterns conflict with traditional tactics of the Go game, which provide human players with new QiGan[1].

In sum, in this paper, we propose to solve distinctive challenges in explaining the Go game. We extend the AND interaction to the OR interaction, penalize unreliable high-order interactions, and compute attributions of common coalitions. Expert human players claim that they have learned new QiGan[1] (novel shape patterns beyond current human knowledge) encoded by the value network.

**Value beyond the Go game.** Discovering inference patterns encoded by a DNN is a generic AI problem. We have also used similar techniques (1) to explain the Gobang game[4] and (2) to explain incorrect patterns used by a DNN for object detection[4]. Please see Appendix M.3 and M.4 for details.

---

[4]Detailed experimental results is shown in Appendix M.3 and M.4 due to the limit of page number.

## 2 EXPLAINING SHAPE PATTERNS OF STONES

### 2.1 PRELIMINARIES: INTERACTIONS

**Definition and computation of the interaction.** In this paper, we use the value network $v$ for the Go game as an example to introduce interactions between different stones encoded by the value network. The value network uses the current state $\boldsymbol{x}$ on the board to estimate the probability $p_{\text{white}}(\boldsymbol{x}) \in [0, 1]$ of white stones winning. To simplify the notation, let us use $\boldsymbol{x} = \{x_1, x_2, ..., x_n\}$ to denote both positions and colors of $n$ stones in the current state. We consider these $n$ stones, including both white and black stones, as input variables[5] of the value network, which are indexed by $N = \{1, 2, ..., n\}$. We set a scalar $v(\boldsymbol{x}) = \log(\frac{p_{\text{white}}(\boldsymbol{x})}{1-p_{\text{white}}(\boldsymbol{x})}) \in \mathbb{R}$ as the advantage of white stones in the game.

In this way, Harsanyi (1963) has proposed a metric $I(S)$, namely the *AND interaction*. We can use the following equation compute the interaction effect $I(S) \in \mathbb{R}$ between stones in $S$, which is encoded by the value network. Given interaction effects $I(S)$ *w.r.t.* all combinations $S \subseteq N$, only those with none-zero effects are taken as valid interactions in $v(x)$.

$$\forall S \subseteq N, \quad I(S) \overset{\text{def}}{=} \sum_{T \subseteq S} (-1)^{|S|-|T|} \cdot v(\boldsymbol{x}_T) \tag{1}$$

where $\boldsymbol{x}_T$ denotes the state when we keep stones in the set $T$ on the board, and remove stones in $N \setminus T$. Thus, $v(\boldsymbol{x}_T)$ measures the advantage score estimated on the masked board state $\boldsymbol{x}_T$.

**Physical meaning and faithfulness guarantee.** *The proven sparsity property and the universal-matching property both clarify the physical meaning of interactions and ensure that interactions faithfully represent primitive inference patterns encoded by the DNN.*

First, Theorem 2.1 shows the ***universal-matching property*** of interactions, *i.e.*, **we can use interactions $S \subseteq N$ to accurately mimic the network outputs on $\boldsymbol{x}_T$, no matter how we randomly mask the input sample $\boldsymbol{x}$.** We can use interactions $S \subseteq N$ to construct a surrogate logical model of AND operations $h(\cdot)$, and the logical model $h(\boldsymbol{x}_T)$ can accurately mimic the network outputs on $\boldsymbol{x}_T$, no matter how we randomly mask the input sample $\boldsymbol{x}$.

Physically, each interaction $S$ in Equation (2) represents a non-linear AND relationship (shape pattern) between stones in $S \subseteq N$, which is equivalently encoded by the neural network $v$. Let us consider the interaction $I(S_2 = \{1, 3, 8\}) \neq 0$ in Figure 1 as an example. If all stones in $S_2$ are placed on the board, then the interaction will make an effect $I(S_2)$ on the advantage score $v(x)$. Thus, we can consider the interaction as an AND relationship $I(S_2) = w_{S_2} \cdot [exist(x_1) \wedge exist(x_3) \wedge exist(x_8)]$ encoded by $v$. The removal of any stones in $S_2$ will remove the effect from $v$.

**Theorem 2.1** (proved by Ren et al. (2023))**.** *Given an input sample $\boldsymbol{x}$, let us construct the following logical model $h(\cdot)$ based on AND interactions. The output score of the DNN on all $2^n$ randomly masked samples $\boldsymbol{x}_T$ w.r.t. $\forall T \subseteq N$ can all be well matched by these interactions.*

$$\forall T \subseteq N, h(\boldsymbol{x}_T) = v(\boldsymbol{x}_T).$$
$$h(\boldsymbol{x}_T) = v(\emptyset) + \sum_{S \subseteq N, S \neq \emptyset} I(S) \cdot \mathbb{1}\left(\begin{smallmatrix} \boldsymbol{x}_T \ triggers \\ the \ AND \ relation \ S \end{smallmatrix}\right) = v(\emptyset) + \sum_{S \subseteq T, S \neq \emptyset} I(S) \tag{2}$$

In addition, the ***sparsity property*** of interactions has been observed by Li & Zhang (2023) and proven by Ren et al. (2024a), *i.e.*, most DNNs only encode a small number of interactions[6] between input variables in an input sample. It means that among all $2^n$ different subsets $S_1, S_2, ..., S_{2^n} \subseteq N$, most interactions have negligible effects $I(S_i) \approx 0$, and only a few interactions in the set $\Omega_{\text{salient}}$[7] have large effects, *s.t.* $|\Omega_{\text{salient}}| \ll 2^n$. Combining Theorem 2.1, $v(\boldsymbol{x}_T)$ is mainly determined by the small number of salient interactions $S \in \Omega_{\text{salient}}$[7], so these salient interactions can be considered as primitive inference patterns encoded by the value network, namely **interaction primitives**.

---

[5]Although the input of the value network is a tensor (Silver et al., 2016), we use $\boldsymbol{x} = \{x_1, x_2, ..., x_n\}$ to denote the input.

[6] Ren et al. (2024a) have partially proved the sparsity of interactions under three common conditions. Although there is no strict way to directly examine whether the network inference on a given input sample satisfy the three conditions, experimental results in Figure 2 and Figure 6 (b) verify the sparsity of interactions encoded by the KataGo model. Please see Appendix D for more detailed introductions of common conditions.

[7]Unlike traditional sparsity, Ren et al. (2024a) define the sparsity as many non-salient interactions, instead of many zero values. We set $\xi = 0.15 \cdot \max_S |I(S)|$ to select salient interactions, $\Omega_{\text{salient}} = \{S : |I(S)| > \xi\}$.

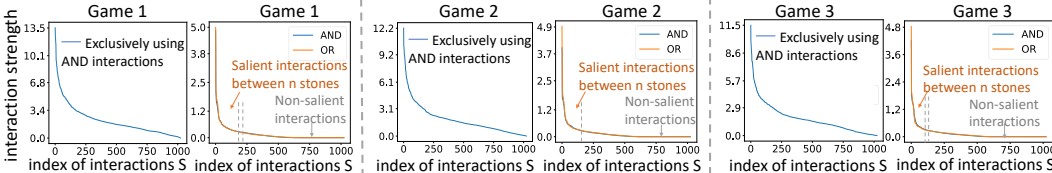

Figure 2: Comparison of the sparsity of interactions[7]. We sort strength of all interaction effects in a descending order. Interaction explanations generated by using both AND and OR interactions are sparser than those generated by using exclusively AND interactions.

The sparsity of interactions universally appears on diverse DNNs with fully different architectures (MLPs, CNNs, Transformers, RNNs) for different tasks (CV, NLP, 3D point clouds). Note that unlike traditional techniques (Tibshirani, 1996), the sparsity here does not mean many exactly zero values but rather many negligible interactions, subject to $|I(S)| < \xi^7$, where $\xi^7$ is a tiny threshold. Please see (Ren et al., 2024a) for detailed proof.

**Complexity of the interaction primitive.** The complexity of an interaction primitive $S$ is defined as the order of the interaction, *i.e.*, the number of stones in $S$, order$(S) = |S|$. An interaction primitive of a higher order represents a more complex interaction with more stones.

## 2.2 EXTENSION TO OR INTERACTIONS[8]

We aim to extract interaction primitives from the value network, and use them as shape patterns that determine the winning probability. It is widely believed that the Go game applies much more complex logic than other applications (Shin et al., 2021). Thus, the network for the Go game is supposed to encode both AND relationships and OR relationships between stones.

However, we find that Theorem 2.2 shows that exclusively using AND interactions to explain an OR relationship will significantly complicate the explanation, *i.e.*, the OR relationship can be represented as an exponential number of specific AND operations[9].

**Theorem 2.2** (proved in Appendix I). *Given an input sample $x = [x_1, x_2, ..., x_n]^\top$, where each input variable $x_{k_j} \in \{0, 1\}$ is binary to represent the presence or absence state of the variable. Let the target function $v(x)$ encode a $m$-order OR relationship, i.e., $v(x) = x_{k_1} \vee x_{k_2} \vee ... \vee x_{k_m}$. Then, the function $v(x)$ would activate $2^m - 1$ non-zero AND interactions, i.e. for all non-empty subset, $\emptyset \neq S \subseteq \{x_{k_1}, x_{k_2}, ..., x_{k_m}\}$, $I_{and}(S|x) = (-1)^{|S|-1} \neq 0$.*

**Definition of OR interactions.** Therefore, we extend AND interactions in Equation (1) to OR interactions. Figure 2 shows that when we simultaneously use AND and OR interactions, we usually obtain sparser explanation than exclusively using AND interactions. Specifically, we decompose the advantage score $v(\boldsymbol{x}_T)$ into a component $v_{and}(\boldsymbol{x}_T)$ for AND interactions, and a component $v_{or}(\boldsymbol{x}_T)$ for OR interactions, *i.e.*, we set $\forall T \subseteq N, T \neq \emptyset, v_{and}(\boldsymbol{x}_T) = \frac{1}{2}v(\boldsymbol{x}_T) + p_T, v_{or}(\boldsymbol{x}_T) = \frac{1}{2}v(\boldsymbol{x}_T) - p_T$, which satisfy $v(\boldsymbol{x}_T) = v_{and}(\boldsymbol{x}_T) + v_{or}(\boldsymbol{x}_T)$. $p_T \in \mathbb{R}$ denotes a learnable bias term. The component $v_{and}(\boldsymbol{x}_T)$ is explained by AND interactions $I_{and}(S) \stackrel{\text{def}}{=} \sum_{T \subseteq S}(-1)^{|S|-|T|} \cdot v_{and}(\boldsymbol{x}_T)$ in Equation (1), subject to $v_{and}(\boldsymbol{x}_T) = v(\emptyset) + \sum_{S \subseteq T, S \neq \emptyset} I_{and}(S)$. The component $v_{or}(\boldsymbol{x}_T)$ is explained by the following OR interactions[10], subject to $v_{or}(\boldsymbol{x}_T) = \sum_{S \cap T \neq \emptyset, S \neq \emptyset} I_{or}(S)$.

$$\forall S \subseteq N, \quad S \neq \emptyset, \quad I_{or}(S) \stackrel{\text{def}}{=} -\sum_{T \subseteq S}(-1)^{|S|-|T|}v_{or}(\boldsymbol{x}_{N \setminus T}) \tag{3}$$

Above settings for OR interactions are justified by the universal matching property (introduced in Theorem 2.4 and experimentally verified in Figure 6(a)) and sparsity property (verified in Figure 2).

---

[8]We put the pseudo-code in Appendix A.

[9]For example, we represent the effect of an AND interaction $S = \{1, 2, 3\}$ as $I_{and}(S) = w_S^{\text{and}} \cdot [exist(x_1) \wedge exist(x_2) \wedge exist(x_3)]$. In comparison, the effect of an OR interaction $S = \{1, 2, 3\}$ is represented as $I_{or}(S) = w_S^{\text{or}} \cdot [exist(x_1) \vee exist(x_2) \vee exist(x_3)] = w_S^{\text{or}} \cdot \{\neg[(\neg exist(x_1)) \wedge (\neg exist(x_2)) \wedge (\neg exist(x_3))]\}$, where $\vee$ represents the binary logical operation "OR." The Boolean function $exist(\cdot) = false$ when the stone is removed.

[10]Please see Appendix F.1 for the proof.

**How to understand OR interactions.** Just like the AND interaction, an OR interaction $S$ is measured to reflect the strength of an OR relationship between stones in the set $S$ encoded by the model $v_{\text{or}}$. If any stone in $S$ appears on the board, then the OR interaction $S$ makes an effect $I_{\text{or}}(S)$ on the score $v_{\text{or}}(\boldsymbol{x})$. Only when all stones in $S$ are removed from the board, the effect $I_{\text{or}}(S)$ is removed.

**Learning of AND-OR interactions**[11]**.** Therefore, the core task of extracting AND-OR interactions is to learn parameters $\{p_T\}_{T \subseteq N}$ to determine the decomposition of the AND-interaction component $v_{\text{and}}(\boldsymbol{x}_T)$ and the OR-interaction component $v_{\text{or}}(\boldsymbol{x}_T)$. However, the core challenge is that the extraction of AND interactions and OR interactions is neither stable, nor unique. Theorem F.1 in Appendix F.2 proves that small unexplainable noises in the network output usually significantly enlarge the strength of high-order interactions. To overcome this problem, we slightly revise the original network output $v(\boldsymbol{x}_T)$ as $v(\boldsymbol{x}_T) + q_T$, where $q_T \in \mathbb{R}$ is a small scalar contained within a small range, $|q_T| < \tau$[12]. The parameters $\{q_T\}_{T \subseteq N}$ are learned to represent unavoidable noises in the network output, which cannot be reasonably explained by AND interactions or OR interactions. We use the following loss function to learn a sparse decomposition of AND and OR interactions, according to the Occam's Razor.

$$\min_{\{p_T\}_{T \subseteq N, T \neq \emptyset}, \{q_T : |q_T| < \tau\}_{T \subseteq N, T \neq \emptyset}} \|\boldsymbol{I}_{\text{and}}\|_1 + \|\boldsymbol{I}_{\text{or}}\|_1 \tag{4}$$

where $\|\cdot\|_1$ represents L-1 norm. $\boldsymbol{I}_{\text{and}} = [I_{\text{and}}(S_1), I_{\text{and}}(S_2), ..., I_{\text{and}}(S_{2^n})]^\top$ and $\boldsymbol{I}_{\text{or}} = [I_{\text{or}}(S_1), I_{\text{or}}(S_2), ..., I_{\text{or}}(S_{2^n})]^\top$ denote all AND interactions and all OR interactions, respectively. AND interactions $\{I_{\text{and}}(S)\}_{S \subseteq N}$ are computed by setting $v_{\text{and}}(\boldsymbol{x}_T) = \frac{1}{2} \cdot [v(\boldsymbol{x}_T) + q_T] + p_T$ according to Equation (1). OR interactions $\{I_{\text{or}}(S)\}_{S \subseteq N}$ are computed by setting $v_{\text{or}}(\boldsymbol{x}_T) = \frac{1}{2} \cdot [v(\boldsymbol{x}_T) + q_T] - p_T$ in Equation (3).

Experimental results in Figure 2 have demonstrated the effectiveness of the proposed method.

**Mathematical properties of OR interactions.** Notably, extending AND interactions to OR interactions does not hurt theoretical solidness of the interaction metric. We find that OR interactions also satisfy desirable properties, just like AND interactions. Theorem 2.3 proves that an OR interaction can be considered as a specific AND interaction[9]. Theorem 2.4 proves that we can use AND-OR interactions to explain the DNN's outputs on all $2^n$ randomly masked samples $\{\boldsymbol{x}_T\}_{T \subseteq N}$. Theorem 2.5 proves that we can use AND-OR interactions to explain the Shapley value (Shapley, 2016).

**Theorem 2.3** (proved in Appendix E). *The OR interaction effect between a set $S$ of stones, $I_{or}(S)$ based on $v(\boldsymbol{x}_T)$, can be computed as a specific AND interaction effect $I'_{and}(S)$ based on the dual function $v'(\boldsymbol{x}_T)$. For $v'(\boldsymbol{x}_T)$, original present stones in $T$ (based on $v(\boldsymbol{x}_T)$) are considered as being removed, and original removed stones in $N \setminus T$ (based on $v(\boldsymbol{x}_T)$) are considered as being present[9].*

**Theorem 2.4** (**universal-matching property**, proved in Appendix G). *Let the input sample $\boldsymbol{x}$ be randomly masked. There are $2^n$ masked samples $\{\boldsymbol{x}_T\}$ w.r.t. $2^n$ subsets $T \subseteq N$. The output score on any masked sample $\boldsymbol{x}_T$ can be represented as the sum of effects of AND-OR interactions.*

$$\forall T \subseteq N, v(\boldsymbol{x}_T) = v(\boldsymbol{x}_\emptyset) + \sum_{S \subseteq N} I_{and}(S) \cdot \mathbb{1}\left(\begin{smallmatrix} \boldsymbol{x}_T \ triggers \\ the \ AND \ relation \ S \end{smallmatrix}\right) + \sum_{S \subseteq N} I_{or}(S) \cdot \mathbb{1}\left(\begin{smallmatrix} \boldsymbol{x}_T \ triggers \\ the \ OR \ relation \ S \end{smallmatrix}\right)$$

$$= v(\boldsymbol{x}_\emptyset) + \sum_{S \subseteq T, S \neq \emptyset} I_{and}(S) + \sum_{S \cap T \neq \emptyset, S \neq \emptyset} I_{or}(S)$$

**Theorem 2.5** (proven in Appendix H). *The Shapley value $\phi(i)$ of each input variable $i \in N$ can be explained as a re-allocation of AND-OR interactions, i.e., $\phi(i) = \sum_{S \subseteq N, S \ni i} \frac{1}{|S|} I_{and}(S) + \sum_{S \subseteq N, S \ni i} \frac{1}{|S|} I_{or}(S)$.*

**Experimental verification of the sparsity of interactions.** Although Ren et al. (2024a) have proved that a DNN usually only encodes a small number of interactions for inference under some common conditions[6], it is still a challenge to strictly examine whether the value network fully satisfies these conditions. Although Theorem 2.3 has proved that OR interaction can be considered as a specific AND interaction, in real applications, we still need to verify the sparsity[7] of interactions encoded by the value network for the Go game.

Therefore, we experimentally examine the KataGo v.1.13.0 (Wu, 2019), which is a free open-source neural network for the Go game and has defeated human players. Specifically, we use KataGo to generate a board state by letting KataGo take turns to play $\frac{m}{2}$ moves of black stones and $\frac{m}{2}$ moves of white stones. Note that accurately computing the interactions of all $m$ stones is an NP-hard problem.

---

[11]Please see Algorithm 1 for the pseudo code.

[12]Please see Appendix L.3 for more details about setting the small threshold $\tau$.

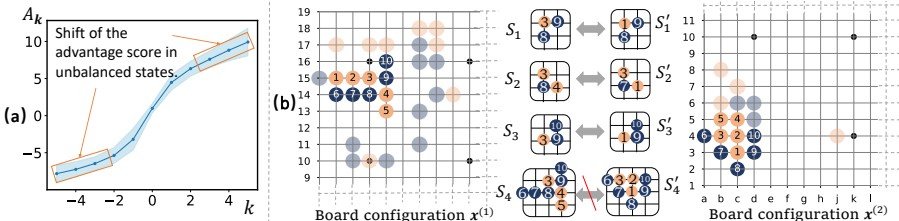

Figure 3: (a) Value shifting over different orders. We show the average advantage score $A_k = \mathbb{E}_{\boldsymbol{x}}\mathbb{E}_{T\subseteq N:\Delta n(T)=k}\log(\frac{p_{\text{white}}(\boldsymbol{x}_T)}{1-p_{\text{white}}(\boldsymbol{x}_T)})$ over all masked states $\boldsymbol{x}_T$ with the same unbalance level $k$. The average advantage score $A_k$ is saturated when $|k|$ is large. (b) Compared to high-order interactions $S_4$, low-order interactions $S_1, S_2, S_3$ from board $\boldsymbol{x}^{(1)}$ are easier to be transferred to another board $\boldsymbol{x}^{(2)}$.

We let *the professional human Go player* to select[13] $n = 10$ stones ($n \le m$), including $\frac{n}{2}$ white stones and $\frac{n}{2}$ black stones, so that we can accurately compute interactions between these $n$ selected stones. All other $(m - n)$ stones are considered as constant background without being masked, and their interactions are not computed. Figure 2 shows the strength $|I(S)|$ of all AND interactions and OR interactions in a descending order. It shows that only a few interactions have salient effects, 80%-85% interactions have negligible effects[7]. The verification of the sparsity of interactions indicates that we can use these sparse interactions as faithful primitive shape patterns encoded by the value network.

## 2.3 How to obtain simple and transferable shape patterns[8]

**Can shape patterns extracted from a board be transferred to other boards?** The transferability of interaction primitives is crucial for the explanation of the Go game.

First, Zhou et al. (2024) have found that high-order interactions are more likely to represent non-generalizable abnormal interactions, while low-order interactions are usually more generalizable, *i.e.*, low-order interactions usually represent common patterns that frequently appear in different games. For example, as Figure 3 (b) shows, 3-order interaction primitives $S_1, S_2, S_3$ extracted from the board state $\boldsymbol{x}^{(1)}$ can be transferred to the board state $\boldsymbol{x}^{(2)}$. However, 8-order interaction primitive $S_4$ extracted from the board state $\boldsymbol{x}^{(1)}$ cannot be transferred to the board state $\boldsymbol{x}^{(2)}$.

Second, Ren et al. (2024b) have proven that high-order interactions lead the DNN to over-fit and are usually penalized in the training process of a DNN, because they represent unstable noise patterns.

**Boosting transferability[14] of interaction primitives.** Therefore, in this subsection, we propose a series of techniques to boost the transferability of interaction primitives. To this end, we find a problem that the KataGo model extracts some high-order interaction primitives (see Figure 4). We prove that the emergence of high-order interactions is caused by the value shift (or called the *saturation problem*) of the value network in Appendix J. *I.e.*, most training data for the value network are usually biased/shifted to states with similar numbers of white stones and black stones, because in real games, the board always contains similar numbers of white stones and black stones. We use $\Delta n(T) = n_{\text{white}}(T) - n_{\text{black}}(T) \in \{-\frac{n}{2}, -\frac{n}{2}+1, ..., \frac{n}{2}\}$ to measure the unbalance level of the masked state $\boldsymbol{x}_T$, where $n_{\text{white}}(T)$ and $n_{\text{black}}(T)$ denote the number of white stones and that of black stones on $\boldsymbol{x}_T$, respectively. As Figure3 (a) shows, we compute the average advantage score $A_k = \mathbb{E}_{\boldsymbol{x}}\mathbb{E}_{T\subseteq N:\Delta n(T)=k}\log(\frac{p_{\text{white}}(\boldsymbol{x}_T)}{1-p_{\text{white}}(\boldsymbol{x}_T)})$ over all masked states $\boldsymbol{x}_T$ with the same unbalance level $k \in \{-\frac{n}{2}, -\frac{n}{2}+1, ..., \frac{n}{2}\}$. The average advantage score $A_k$ is not roughly linear with $k$, but is saturated when $|k|$ is large. Appendix J further shows that the value shift causes more high-order interactions.

**Solution[11].** Therefore, we propose to revise the advantage score $v(\boldsymbol{x}_T)$ in Equation (1) to remove the value shift and thereby weaken high-order interactions. Specifically, we remove the value shift by setting $u(\boldsymbol{x}_T) = v(\boldsymbol{x}_T) - a_k$. Given a masked state $\boldsymbol{x}_T$, we compute its unbalance level $k = \Delta n(T) = n_{\text{white}}(T) - n_{\text{black}}(T) \in \{-\frac{n}{2}, -\frac{n}{2}+1, ..., \frac{n}{2}\}$. $a_k$ is initialized as the average advantage score $A_k$. We extend the loss function in Equation (4) as follows to learn the parameters $\boldsymbol{a} =$

---

[13]We admit that the selection of stones affects a bit the interactions, but empirically it does not dramatically change the conclusion. *I.e.*, the sparsity of interactions can be ensured no matter how we select stones. In addition, professional human player ensures that the selected stones do have potential correlations.

[14]Please see Figure 7 in Appendix M.1 for experimental verification of the transferability of the new interactions proposed in this paper.

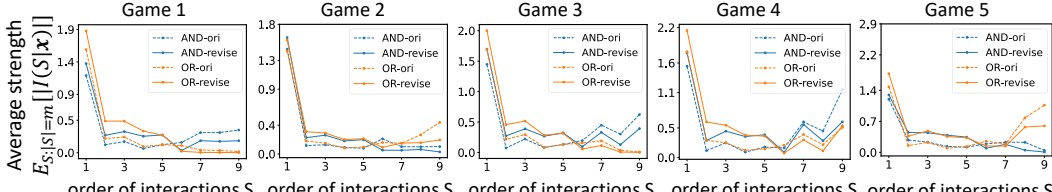

Figure 4: Effectiveness of reducing high-order interactions. We show average strength of effects for interactions of different orders. For different games, the revised method extracts weaker high-order interactions than the original method.

$$\{a_{-\frac{n}{2}}, a_{-\frac{n}{2}+1}, ..., a_{\frac{n}{2}}\}, \{p_T\}_{T \subseteq N}, \{q_T\}_{T \subseteq N}.$$

$$\min_{\boldsymbol{a}, \{p_T\}_{T \subseteq N, T \neq \emptyset}, \{q_T : |q_T| < \tau\}_{T \subseteq N, T \neq \emptyset}} \|\boldsymbol{I}_{\text{and}}\|_1 + \|\boldsymbol{I}_{\text{or}}\|_1 \tag{5}$$

We learn parameters to obtain the sparse decomposition of AND interactions $\boldsymbol{I}_{\text{and}} = [I_{\text{and}}(S_1), I_{\text{and}}(S_2), ..., I_{\text{and}}(S_{2^n})]^\top$ and OR interactions $\boldsymbol{I}_{\text{or}} = [I_{\text{or}}(S_1), I_{\text{or}}(S_2), ..., I_{\text{or}}(S_{2^n})]^\top$. AND interactions $\{I_{\text{and}}(S)\}_{S \subseteq N}$ are computed by setting $v_{\text{and}}(\boldsymbol{x}_T) = \frac{1}{2} \cdot [u(\boldsymbol{x}_T) + q_T] + p_T$, according to the paradigm of Equation (1). OR interactions $\{I_{\text{or}}(S)\}_{S \subseteq N}$ are computed by setting $v_{\text{or}}(\boldsymbol{x}_T) = \frac{1}{2} \cdot [u(\boldsymbol{x}_T) + q_T] - p_T$ in Equation (3). Appendix L.3 shows details about setting the threshold $\tau$. *Experimental results in Figure 4 have verified the effectiveness of our method in penalizing high-order interactions.*

**Penalizing high-order interactions.** Besides, we can add a loss to Equation (5) to penalize high-order interactions, *i.e.*, $Loss = \|\boldsymbol{I}_{\text{and}}\|_1 + \|\boldsymbol{I}_{\text{or}}\|_1 + r \cdot (\|\boldsymbol{I}_{\text{and}}^{\text{high}}\|_1 + \|\boldsymbol{I}_{\text{or}}^{\text{high}}\|_1)$, where $\boldsymbol{I}_{\text{and}}^{\text{high}}$ and $\boldsymbol{I}_{\text{or}}^{\text{high}}$ denote the shorter vectors corresponding to high-order AND interactions and OR interactions, respectively. We empirically set $\boldsymbol{I}_{\text{and}}^{\text{high}}$ as a 386-dimensional vector that corresponds to all interactions between the 6-th order to the 10-th order, according the suggestions of professional human players[15]. We set $r = 5.0$ to boost the penalty of high-order interactions. *Experimental results in Figure 4 have verified the effectiveness of our method in penalizing high-order interactions.*

**High-order shape patterns $\neq$ global shape patterns.** Human Go players typically use global shape patterns to play the Go game, rather than use high-order shape patterns. A global shape means that stones in the shape locate across different regions of the board, but it does not require that all stones be contained within the shape. In contrast, the high-order shape means most of the $n$ stones are contained within the shape, rather than require all selected stones to locate in different regions.

**Experiments.** (1) We conduct the first experiment to check whether above methods can reduce the complexity (order) of the extracted interactions, compared with the original methods in Equation (4). Specifically, we follow experimental settings in Section 2.2 to generate a board, and compute AND-OR interactions between selected $n$ stones. Then, we compute the average strength of AND-OR interactions of different orders, $\mathbb{E}_{S:|S|=m}[|I_{\text{and}}(S)|]$ and $\mathbb{E}_{S:|S|=m}[|I_{\text{or}}(S)|]$, respectively. Figure 4 shows the average strength of interaction effects. For both AND interactions and OR interactions, we observe that the revised method extracts much weaker high-order interactions than the original method in Equation (4). It verifies that our method can reduce the complexity of interactions. (2) We conduct the second experiment to show that our method successfully ensure a high transferability of interactions between stones through different boards. Please see Figure 7 in Appendix M.1 for details.

## 2.4 DISCOVERING NOVEL SHAPE PATTERNS[8]

In the above section, we have extracted sparse and simple interaction primitives from the value network. In this section, we aim to discover novel shape patterns from these interaction primitives beyond human understanding of the Go game. Please see Algorithm 1 for the pseudo code.

We have examined the sparsity of interaction primitives in experiments. We can usually extract about 100–250 interaction primitives to explain the output of a single board state. However, the number of primitives is still too large, and we need a more efficient way to discover novel shape patterns. Therefore, we visualize all primitives, and then identify some specific combinations of stones that

---

[15]This is a fully empirical setting. We set the sixth order as the high order, because it is difficult for professional human players to find any meaningful shape patterns from interactions of more than the 6-th order. Anyway, users can also set other orders when they extend this method to explain other games.

frequently appear in different interaction primitives. We refer to these combinations as "*common coalitions*." For example, in Figure 1, given a board state $\boldsymbol{x}$ with $n$ stones, $N = \{1, 2, ..., n\}$, we can extract some salient interactions from the board state $\boldsymbol{x}$, such as $S_1 = \{1, 3, 4, 5, 8\}, S_2 = \{1, 3, 8\}$, $S_3 = \{1, 3, 8, 9\}$, *etc*. The coalition $T = S_1 \cap S_2 \cap S_3 = \{1, 3, 8\}$ participates in different interactions $T \subseteq S_1, S_2, S_3$. We can consider this coalition $T$ as a *shape pattern* encoded by the value network.

Therefore, we further compute the attribution $\varphi(T)$ of each coalition $T$ to the advantage score $v(\boldsymbol{x})$ estimated by the value network. In this way, $\varphi(T) > 0$ means that the shape pattern of the coalition $T$ tends to enhance the advantage of white stones. In comparison, $\varphi(T) < 0$ means that the shape pattern of the coalition $T$ tends to decrease the advantage score. $\varphi(T) \approx 0$ means that although the coalition $T$ is well modeled by the value network, the coalition $T$ has contradictory effects when it appears in different interactions, thereby not making a significant effect on the advantage score.

There are many attribution methods (Lundberg & Lee, 2017; Selvaraju et al., 2017; Zhou et al., 2016; Zintgraf et al., 2017) to estimate the attribution/importance score of different input variables of an AI model, *e.g.*, estimating attributions of different image patches to the image-classification score, or the attributions of different tokens in natural language processing. However, there is no a widely accepted method to estimate the attribution of a coalition of input variables, because most attribution methods cannot generate self-consistent attribution values[16]. Therefore, we apply the attribution method in (Xinhao Zheng, 2023) to define the attribution of a coalition $T$. This method extends the theory of the Shapley value and well explains the above inconsistency problem. Specifically, the attribution score $\varphi(T)$ of the coalition $T$ is formulated as the weighted sum of effects of AND-OR interactions.

$$\varphi(T) = \sum\nolimits_{S \supseteq T} (|T|/|S|) \cdot [I_{\text{and}}(S) + I_{\text{or}}(S)] \tag{6}$$

$$\varphi(T) - \sum\nolimits_{i \in T} \phi(i) = \sum\nolimits_{S \cap T \neq \emptyset, S \cap T \neq T} (|S \cap T|/|S|) \cdot [I_{\text{and}}(S) + I_{\text{or}}(S)] \tag{7}$$

Let there be some AND interactions and OR interactions containing the coalition $T$. Then, Equation (6) shows that for each interaction $S \supseteq T$ containing the coalition $T$, we must allocate a ratio $\frac{|T|}{|S|}$ of its interaction effect as a numerical component of $\varphi(T)$. **In addition, Appendix K shows a list of desirable theorems and properties of the attribution of the coalition defined in Equation (6), which theoretically guarantee the faithfulness of the attribution metric $\varphi(T)$.** For example, Equation (7) explains $\varphi(T) - \sum_{i \in T} \phi(i)$, *i.e.*, the difference between the coalition's attribution $\varphi(T)$ and the sum of Shapley values $\phi(i)$ for all input variables $i$ in $T$, where the input variable $i$'s Shapley value (defined as $\phi(i) = \sum_{S \subseteq N \setminus \{i\}} \frac{|S|!(n-|S|-1)!}{n!}[u(\boldsymbol{x}_{S \cup i}) - u(\boldsymbol{x}_S)]$) can also be proved/explained as $\phi(i) = \sum_{S \ni i} \frac{1}{|S|}[I_{\text{and}}(S) + I_{\text{or}}(S)]$. The difference comes from those interactions that only contain partial variables in $T$, not all variables in $T$. Please see Appendix K for more theorems.

*Experiments.* Given a board state, we extract interaction primitives encoded by the value network, *i.e.*, $\{S : |I(S)| > \xi\}$, where $\xi = 0.15 \cdot \max_S |I(S)|$. Then, we manually annotate 50 coalitions based on the guidance from professional human Go players. Figure 5 visualizes sixteen coalitions selected from four game states. We compute the attribution of contextual stones to a target coalition $S$, and visualize attribution values of contextual stones in Figure 8 of Appendix L.4. Please see Appendix L.4 for more details about the computation and visualization of the interaction context's attribution.

## 2.5 HUMAN INTERPRETATION OF THE CLASSIC SHAPE PATTERNS (QiGan[1])

In order to explain shape patterns (coalitions) encoded by the value network, we collaborate with professional human Go player[17]. Based on Figure 5, they find both shape patterns that fit QiGan[1] of human players and shape patterns that conflict with human understandings (QiGan[1]).

**Cases that fit traditional human understandings (traditional QiGan[1]). For the Game 1** in Figure 5 (1.a - 1.d), $\varphi(\{1, 2, 3, 8\}) < \varphi(\{2, 3, 8\})$ and $\varphi(\{1, 2, 3, 7\}) < \varphi(\{2, 3, 7\})$. It means that

---

[16]We use the following example to introduce the inconsistency problem. We can simply consider a coalition $T$ (*e.g.*, $T = \{1, 2, 3\}$) of input variables as a singleton variable $[T]$, then we have a total of $n - 2$ input variables in $N' = \{[T], 4, 5, ..., n\}$. Let $\varphi([T])$ denote the attribution of $[T]$ computed on the new partition $N'$ of the $n - 2$ variables. Alternatively, we can also consider $x_1, x_2, x_3$ as three individual variables, and compute their attributions $\varphi(1), \varphi(2), \varphi(3)$ given the original partition of input variables $N = \{1, 2, ..., n\}$. However, for most attribution methods, $\varphi([T]) \neq \varphi(1) + \varphi(2) + \varphi(3)$. This is the inconsistency problem of attributions.

[17]During the review phase, the Go players are anonymous, because they are also authors.

when the white stone $x_1$ participates in the combination of white stones $x_2, x_3$, the advantage of white stones become lower, *i.e.*, the stone $x_1$ is a low-value move. Go players also consider that the effect of the combination of white stones $x_1, x_2, x_3$ is low. **For the Game 2** in Figure 5 (2.a, 2.b), $\varphi(\{1, 2, 8\}) < \varphi(\{1, 5, 8\})$ means that the value network considers that the white stone $x_2$ has lower value than $x_5$. Go players consider that in this game state, the white stone $x_5$ protects the white stones $x_1, x_2, x_3, x_4$, and the white stones $x_1, x_3$ attack the black stones $x_6, x_7$, but the white stone $x_2$ has much less value than other stones. **For the Game 3** in Figure 5 (3.a - 3.c), $\varphi(S_1 = \{1, 3, 8\})$ $> \varphi(S_3 = \{1, 2, 3, 8\})$ and $\varphi(S_2 = \{2, 3, 8\}) > \varphi(S_3 = \{1, 2, 3, 8\})$, subject to $S_3 = S_1 \cup S_2$. It means that given the context $S_1$, the stone $x_2$ wastes a move, and given the context $S_2$, the stone $x_1$ wastes a move. Go players also consider that the combination of stones $x_1, x_2$ is of low value.

**Cases that conflict with human understandings (QiGan[1]). For the Game 3** in Figure 5 (3.d), $\varphi(\{6, 7, 8\}) = 1.00$. Go players are confused that the coalition $\{6, 7, 8\}$ is advantageous for white stones. Since there are 3 black stones and no white stones in $\{6, 7, 8\}$, they believe this coalition is advantageous for black stones. **For the Game 4** in Figure 5 (4.a, 4.b), $\varphi(\{1, 2, 3, 9\}) = -1.71$ and $\varphi(\{1, 2, 3, 4\}) = 1.34$. It means that the coalition $\{1, 2, 3, 4\}$ is advantageous for white stones, and the coalition $\{1, 2, 3, 9\}$ is advantageous for black stones, which are contrary to Go players' QiGan[1].

*Please see Appendix M.2 and Figure 9 for analysis on more boards.*

### 2.6 Do we really discover novel shape patterns or just wrong patterns?

An core issue is distinguishing whether extracted shape patterns that conflict with human understandings represent novel shape patterns or incorrect explanations. Thus, we conduct three examinations to check if the interactions act as primitive inference patterns encoded by the value network.

**First, let us examine the sparsity[7] of interactions, *i.e.*, checking whether the revised method can still provide a few representative shape patterns, instead of many noisy patterns.** We follow experimental settings in Section 2.2 to generate 50 game states, and visualize the strength of all AND interactions and all OR interactions of all these 50 game states in a descending order. Figure 6 (b) shows that only a few interactions have salient effects, and more than $90\%$ interactions have small effects, which verifies the sparsity[7] of interactions extracted by the revised method.

**Second, let us examine the universal-matching property of the revised method.** Theoretically, the AND-OR interactions extracted by our revised method still satisfy the universal matching property in Theorem 2.4. To verify this, we conduct experiments on a given board state $\boldsymbol{x}$, and we check if the extracted AND-OR interactions can accurately mimic the network outputs $u(\boldsymbol{x}_T)$ on all $2^n$ randomly masked board states $\{\boldsymbol{x}_T\}_{T \subseteq N}$. To this end, for each masked board states $\boldsymbol{x}_T$, we measure the approximation error $\Delta u_T = |u^{\text{real}}(\boldsymbol{x}_T) - u^{\text{approx}}(\boldsymbol{x}_T)|$ of using AND-OR interactions to mimic the real output $u^{\text{real}}(\boldsymbol{x}_T)$, where $u^{\text{approx}}(\boldsymbol{x}_T) = u(\boldsymbol{x}_\emptyset) + \sum_{S \subseteq T, S \neq \emptyset} I_{\text{and}}(S) + \sum_{S \cap T \neq \emptyset, S \neq \emptyset} I_{\text{or}}(S)$ represents the score approximated by AND-OR interactions according to Theorem 2.4. In Figure 6 (a), the solid curve shows the real network outputs on all $2^n$ randomly masked board states in an ascending order. The shade area shows the smoothed approximation error, which is computed by averaging approximation errors of neighboring 50 masked board states. Figure 6 (a) shows that the approximated outputs $u^{\text{approx}}(\boldsymbol{x}_T)$ can well match with the real outputs $u^{\text{real}}(\boldsymbol{x}_T)$ over different randomly masked states. It means that the output of the value network can be explained as AND-OR interactions.

**Third, experiments in Appendix M.1 shows the transferability of shape patterns, *i.e.*, shape patterns extracted from a board can also explain $u(\boldsymbol{x}_T)$ in another board.**

## 3 Conclusion and discussions

In this paper, we extract sparse interactions between stones memorized by the value network for the game of Go. Then, we examine both the fitness and conflicts between the automatically extracted shape patterns and conventional human understanding of the game of Go, so as to help human players learn novel shape patterns from the value network as the new QiGan[1] to play the Go game. We collaborate with professional human Go players to analyze the QiGan[1] that is automatically extracted from the value network. Note that our explanation method is a generic method of disentangling inference patterns encoded by the DNN. We also show how to use interactions to explain the Gobang game and to debug representation flaws for object detection in Appendices M.3 and M.4.

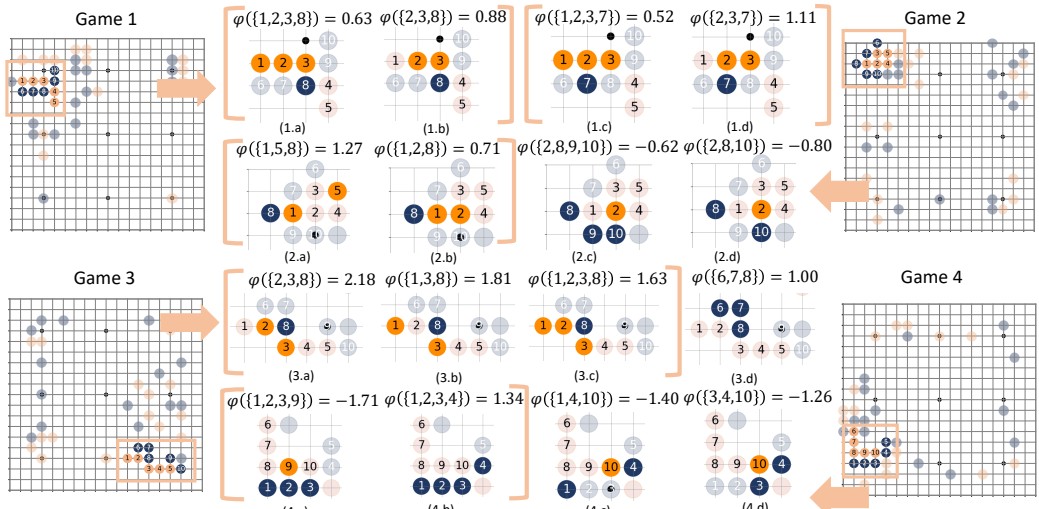

Figure 5: Estimated attributions of different coalitions (shape patterns).

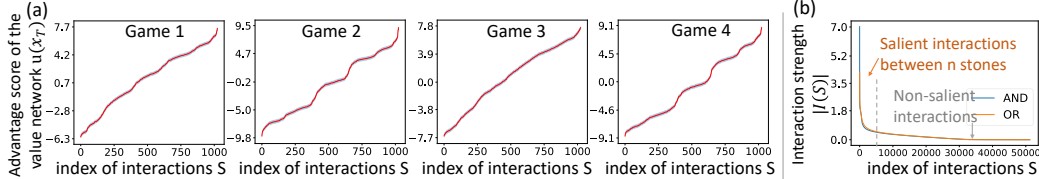

Figure 6: (a) Examination of using the revised interactions to mimic the output of the value network. Outputs of the value network on all $2^n$ masked board states $u(\boldsymbol{x}_T)$ (the red full line) are arranged in ascending order. The height of the blue shade represents the smoothed approximation error, computed by averaging the approximation errors $\Delta u_T = |u^{\text{real}}(\boldsymbol{x}_T) - u^{\text{approx}}(\boldsymbol{x}_T)|$ of neighboring 50 masked states. (b) Sparsity[7] of the revised interactions. We show strength of all revised AND and OR interactions of 50 games in a descending order. Only a few interactions have salient effects.

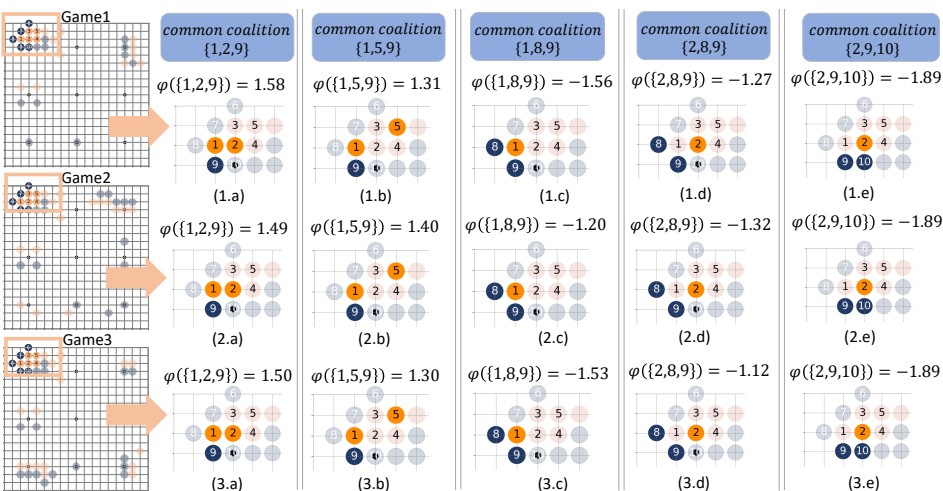

Figure 7: Transferability of interactions over different boards (with different contextual stones). Shape patterns extracted from one board can also explain the network outputs on other boards, *i.e.*, the same interaction contributes similar interaction effects to the advantage scores of different boards.

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

# A    PSEUDOCODE OF OUR METHODS

To clarify how we explain the shape patterns encoded by the value network, we show the pseudocode of our algorithm.

---

**Algorithm 1** Disentangling the shape patterns encoded by the value network $v(\cdot)$

---

**Input:** a board state $x$, a value network $v(\cdot)$
**Output:** All AND interactions $I_{\text{and}}(S)$ and all OR interactions $I_{\text{or}}(S)$, *w.r.t.* $\forall S \subseteq N$, and the attributions $\varphi(T)$ of each coalition $T$
**Step 1:** initializing parameters.
$\quad \forall k \in \{-\frac{n}{2}, -\frac{n}{2}+1, ..., \frac{n}{2}\}$, set $a_k = \mathbb{E}_{\boldsymbol{x}} \mathbb{E}_{T \subseteq N: \Delta n(T)=k} \log(\frac{p_{\text{white}}(\boldsymbol{x}_T)}{1-p_{\text{white}}(\boldsymbol{x}_T)})$.
$\quad \forall T \subseteq N$, set $p_T = q_T = 0$.
**Step 2:** iteratively updating parameters $\boldsymbol{a} = \{a_{-\frac{n}{2}}, a_{-\frac{n}{2}+1}, ..., a_{\frac{n}{2}}\}, \{p_T\}_{T \subseteq N}, \{q_T\}_{T \subseteq N}$
$\quad \min\limits_{\boldsymbol{a}, \{p_T\}_{T \subseteq N}, \{q_T: |q_T| < \tau\}_{T \subseteq N}} \|\boldsymbol{I}_{\text{and}}\|_1 + \|\boldsymbol{I}_{\text{or}}\|_1$ based on Equation (5).
**Step 3:** determining a set of salient interaction primitives $\Omega_{\text{salient}}$.
$\quad \Omega_{\text{salient}} = \{S : |I(S)| > \xi\}$, where $\xi = 0.15 \cdot \max_S |I(S)|$.
**Step 4:** manually annotating common coalitions.
$\quad$ Manually annotate 50 common coalitions $T$ based on interaction primitives $\Omega_{\text{salient}}$.
**Step 5:** computing coalition attributions.
$\quad$ for each annotated coalition $T$, $\varphi(T) = \sum_{S \supseteq T} \frac{|T|}{|S|} [I_{\text{and}}(S) + I_{\text{or}}(S)]$.

---

# B    RELATED WORK

Many methods have been proposed to visualize the feature/patterns encoded by the DNN (Simonyan et al., 2014; Dosovitskiy & Brox, 2016; Yosinski et al., 2015; Zeiler & Fergus, 2014), or to estimate the attribution/attention on each input variable (Lundberg & Lee, 2017; Selvaraju et al., 2017; Zhou et al., 2016; Zintgraf et al., 2017). However, the complexity of the Go game presents a new requirement, *i.e.*, accurately extracting exact shape patterns encoded by the DNN, instead of providing a vague heatmap of attention. Clarifying exact inference patterns is of significant values in knowledge discovery from DNNs towards various tasks. For example, our method can also be used to discover novel patterns for medical diagnosis.

However, explaining exact shape patterns encoded by a DNN proposes higher requirements for the faithfulness/fidelity of the explanation. The faithfulness is even supposed to be theoretically guaranteed and experimentally verified, beyond a specious understanding. To this end, (1) Ren et al. (2023) and Li & Zhang (2023) have found that a well-trained DNN usually encodes a small number of interactions, and the output score of the DNN on a certain input sample can always be well mimicked by numerical effects of a few salient interactions, no matter how the input sample is randomly masked[6]. (2) Li & Zhang (2023) have further found the considerable transferability of interactions over different samples and over different DNNs. (3) Interaction primitives (the AND interaction) can explain the elementary mechanism of previous explanation metrics, *e.g.*, the Shapley value (Shapley, 2016), the Shapley interaction index (Grabisch & Roubens, 1999), and the Shapley Taylor interaction index (Sundararajan et al., 2020).

Despite of above findings, explaining the DNN for the Go game still proposes new challenges. To this end, we extend the AND interaction to the OR interaction, alleviate high-order interactions caused by the value shift of the advantage score, and compute attributions of common coalitions shared by different interactions, thereby obtaining concise and accurate explanation for shape patterns in the value network.

# C    PROPERTIES FOR THE HARSANYI DIVIDEND

In this paper, we follow Ren et al. (2023) to use the Harsanyi dividend (or Harsanyi interaction) to measure the numerical effect $I(S)$ of the interaction primitive $S$. Ren et al. (2023) have proved that

the Harsanyi dividend satisfied the following properties, including the *efficiency, linearity, dummy, symmetry, anonymity, recursive, interaction distribution properties.*

(1) Efficiency property: The inference score of a well-trained model $v(\boldsymbol{x})$ can be disentangled into the numerical effects of different interaction primitives $I(S), S \subseteq N$, *i.e.*, $v(\boldsymbol{x}) = \sum_{S \subseteq N} I(S)$.

(2) Linearity property: If the inference score of the model $w$ is computed as the sum of the inference score of the model $u$ and the inference score of the model $v$, *i.e.*, $\forall S \subseteq N, w(\boldsymbol{x}_S) = u(\boldsymbol{x}_S) + v(\boldsymbol{x}_S)$, then the interactive effect of $S$ on the model $w$ can be computed as the sum of the interaction effect of $S$ on the model $u$ and that on the model $v$, *i.e.*, $\forall S \subseteq N, I_w(S) = I_u(S) + I_v(S)$.

(3) Dummy property: If the input variable $i$ is a dummy variable, *i.e.*, $\forall S \subseteq N \setminus \{i\}, v(\boldsymbol{x}_{S \cup \{i\}}) = v(\boldsymbol{x}_S) + v(\boldsymbol{x}_{\{i\}})$, then the input variable $i$ has no interaction with other input variables, *i.e.*, $\forall \emptyset \neq S \subseteq N \setminus \{i\}, I(S \cup \{i\}) = 0$.

(4) Symmetry property: If the input variable $i \in N$ and the input variable $j \in N$ cooperate with other input variables in $S \subseteq N \setminus \{i, j\}$ in the same way, *i.e.*, $\forall S \subseteq N \setminus \{i, j\}, v(\boldsymbol{x}_{S \cup \{i\}}) = v(\boldsymbol{x}_{S \cup \{j\}})$, then the input variable $i$ and the input variable $j$ have the same interactive effect, *i.e.*, $\forall S \subseteq N \setminus \{i, j\}, I(S \cup \{i\}) = I(S \cup \{j\})$.

(5) Anonymity property: If a random permutation $\pi$ is added to $N$, then $\forall S \subseteq N, I_v(S) = I_{\pi v}(\pi S)$ is always guaranteed, where the new set of input variables $\pi S$ is defined as $\pi S = \{\pi(i), i \in S\}$, the new model $\pi v$ is defined as $(\pi v)(\boldsymbol{x}_{\pi S}) = v(\boldsymbol{x}_S)$. This suggests that permutation does not change the interactive effects.

(6) Recursive property: The interactive effects can be calculated in a recursive manner. For $\forall i \in N, S \subseteq N \setminus \{i\}$, the interactive effect of $S \cup \{i\}$ can be computed as the difference between the interactive effect of $S$ with the presence of the variable $i$ and the interactive effect of $S$ with the absence of the variable $i$. *I.e.*, $\forall i \in N, S \subseteq N \setminus \{i\}, I(S \cup \{i\}) = I(S|i \text{ is consistently present}) - I(S)$, where $I(S|i \text{ is consistently present}) = \sum_{L \subseteq S}(-1)^{|S|-|L|} v(\boldsymbol{x}_{L \cup \{i\}})$.

(7) Interaction distribution property: This property describes how an interaction function (Sundararajan et al., 2020) distributes interactions. An interaction function $v_T$ parameterized by a context $T$ is defined as follows. $\forall S \subseteq N$, if $T \subseteq S$, then $v_T(\boldsymbol{x}_S) = c$; if not, $v_T(\boldsymbol{x}_S) = 0$. Then, the interactive effects for an interaction function $v_T$ can be computed as, $I(T) = c$, and $\forall S \neq T, I(S) = 0$.

## D COMMON CONDITIONS FOR THE SPARSITY OF INTERACTIONS ENCODED BY A DNN

Ren et al. (2024a) have proved that under some common conditions, a DNN usually only encodes a small number of interactions for inference[6]. *I.e.*, (1) The high-order derivatives of the model output with respect to the input variables are all zero. (2) The AI model can be used on masked/occluded samples, and when the input sample is less masked/occluded, the AI model will yield a higher confidence score on this sample. (3) The confidence score of the AI model on masked/occluded samples does not significantly degrade.

## E PROVING THAT THE OR INTERACTIONS CAN BE CONSIDERED AS A SPECIFIC AND INTERACTION

**Theorem 2.3** *The OR interaction effect between a set $S$ of stones, $I_{or}(S)$ based on $v(\boldsymbol{x}_T)$, can be computed as a specific AND interaction effect $I'_{and}(S)$ based on the dual function $v'(\boldsymbol{x}_T)$. For $v'(\boldsymbol{x}_T)$, original present stones in $T$ (based on $v(\boldsymbol{x}_T)$) are considered as being removed, and original removed stones in $N \setminus T$ (based on $v(\boldsymbol{x}_T)$) are considered as being present.*

• **proof:** The effect $I_{or}(S)$ of an OR interaction $S$ is defined as follows.

$$\forall S \subseteq N, S \neq \emptyset, I_{or}(S) = -\sum_{T \subseteq S}(-1)^{|S|-|T|} v(\boldsymbol{x}_{N \setminus T})$$

Here, $\boldsymbol{x}_{N \setminus T}$ denotes the masked board state where stones in the set $N \setminus T$ are placed on the board, and stones in the set $T$ are removed. We reconsider the definition of the masked board state $\boldsymbol{x}$ as the

definition of $\boldsymbol{x}'$. In comparison, $\boldsymbol{x}'_T$ denotes the masked board state where stones in the set $T$ are removed (based on the definition of $\boldsymbol{x}_T$, stones in the set $T$ are placed on the board), and stones in the set $N \setminus T$ are placed on the board (based on $\boldsymbol{x}_T$, stones in the set $N \setminus T$ are removed).

In this way, $\boldsymbol{x}_{N \setminus T}$ denotes the same board state as $\boldsymbol{x}'_T$. The effect $I_{\mathrm{or}}(S|\boldsymbol{x})$ of an OR interaction based on the definition of $\boldsymbol{x}$ can be reformulated as the effect $I'_{\mathrm{and}}(S|\boldsymbol{x}')$ of an AND interaction based on the definition of $\boldsymbol{x}'$ as follows.

$$
\begin{aligned}
I_{\mathrm{or}}(S|\boldsymbol{x}) &= -\sum_{T \subseteq S} (-1)^{|S|-|T|} v(\boldsymbol{x}_{N \setminus T}), \quad S \neq \emptyset \\
&= -\sum_{T \subseteq S} (-1)^{|S|-|T|} v(\boldsymbol{x}'_T), \quad S \neq \emptyset \\
&= -I'_{\mathrm{and}}(S|\boldsymbol{x}'), \quad S \neq \emptyset
\end{aligned}
$$

Therefore, we consider the OR interaction as a specific AND interaction.

# F    PROOFS RELATED TO OR INTERACTIONS

## F.1    PROVING THAT THE MODEL OUTPUT CAN BE REPRESENTED AS OR INTERACTIONS

According to Appendix E, we reconsider the definition of the masked board state $\boldsymbol{x}_T$ as $\boldsymbol{x}'_T$. $\boldsymbol{x}_T$ denotes the masked board state where stones in the set $T$ are placed on the board, and stones in the set $N \setminus T$ are removed. In comparison, $\boldsymbol{x}'_T$ denotes the masked board state where stones in the set $T$ are removed, and stones in the set $N \setminus T$ are placed on the board.

In this way, the effect of an OR interaction based on the definition of $\boldsymbol{x}$ can be represented as the effect $I'_{\mathrm{and}}(S|\boldsymbol{x}')$ of an AND interaction based on the definition of $\boldsymbol{x}'$.

$$
\begin{aligned}
I_{\mathrm{or}}(S|\boldsymbol{x}) &= w_S^{\mathrm{or}} \cdot [-\prod_{i \in S} \neg exist(x_i)] \\
&= -w_S^{\mathrm{or}} \cdot \prod_{i \in S} \neg exist(x_i) \\
&= -\frac{w_S^{\mathrm{or}}}{w_S^{\mathrm{and}}} \cdot I'_{\mathrm{and}}(S|\boldsymbol{x}')
\end{aligned}
$$

where the function $exist(x_i)$ represents that the stone $x_i$ is placed on the board, the function $\neg exist(x_i)$ represents that the stone $x_i$ is removed from the board.

## F.2    PROVING THAT UNAVOIDABLE NOISES IN NETWORK OUTPUT WILL BE ENLARGED IN INTERACTIONS

Actually, the real data inevitably contains some small noises/variations, such as texture variations and the shape deformation in object classification. Therefore, the network output $v(\boldsymbol{x}_T)$ also contains some unavoidable noises.

**Theorem F.1.** *Let $Var[v(\boldsymbol{x}_T)]$ denote the variance of the network output $v(\boldsymbol{x}_T)$, and let $Var[v(\boldsymbol{x}_{N \setminus T})]$ denote the variance of the network output $v(\boldsymbol{x}_{N \setminus T})$. If we assume that different masked input samples are independent of each other and have no correlation, then we can derive the variance of the AND interaction and the variance of the OR interactions will be enlarged.*

$$
\begin{aligned}
Var[I_{and}(S)] &= Var[\sum_{T \subseteq S} (-1)^{|S|-|T|} v(\boldsymbol{x}_T)] = \sum_{T \subseteq S} Var[v(\boldsymbol{x}_T)] \\
Var[I_{or}(S)] &= Var[-\sum_{T \subseteq S} (-1)^{|S|-|T|} v(\boldsymbol{x}_{N \setminus T})] = \sum_{T \subseteq S} Var[v(\boldsymbol{x}_{N \setminus T})]
\end{aligned}
$$

As Theorem F.1 shows, the variance of the masked input sample will enlarged the variance of the AND interactions and the variance of the OR interactions. Therefore, we prove that unavoidable noises in network output will enlarged in interactions.

# G  PROVING THAT THE NETWORK OUTPUT CAN BE REPRESENTED AS AND-OR INTERACTIONS

**Theorem 2.4.** *Let the input sample $x$ be randomly masked. There are $2^n$ possible masked samples $\{x_T\}$ w.r.t. $2^n$ subsets $T \subseteq N$. The output score on any masked sample $x_T$ can be represented as the sum of effects of both AND interactions and OR interactions.*

$$\forall T \subseteq N, v(x_T) = v(x_\emptyset) + v_{and}(x_T) + v_{or}(x_T) = v(x_\emptyset) + \sum_{S \subseteq T, S \neq \emptyset} I_{and}(S) + \sum_{S \cap T \neq \emptyset, S \neq \emptyset} I_{or}(S)$$

• **proof:** We derive that for all $2^n$ randomly masked sample $x_T$, the output score $v(x_T)$ of the DNN on $x_T$ can be approximated by the sum of effects of AND-OR interactions, *i.e.,* $v(x_T) = v(x_\emptyset) + \sum_{S \subseteq T, S \neq \emptyset} I_{and}(S) + \sum_{S \cap T \neq \emptyset, S \neq \emptyset} I_{or}(S)$

$$\sum_{S \subseteq T} I_{and}(S) = \sum_{S \subseteq T} \sum_{L \subseteq S} (-1)^{|S|-|L|} v_{and}(x_L)$$

$$= \sum_{L \subseteq T} \sum_{S:L \subseteq S \subseteq T} (-1)^{|S|-|L|} v_{and}(x_L)$$

$$= \underbrace{v_{and}(x_T)}_{L=T} + \sum_{L \subseteq T, L \neq T} v_{and}(x_L) \cdot \underbrace{\sum_{m=0}^{|T|-|L|} (-1)^m}_{=0}$$

$$= v_{and}(x_T)$$

$$\sum_{S \cap T \neq \emptyset, S \neq \emptyset} I_{or}(S) = -\sum_{S \cap T \neq \emptyset, S \neq \emptyset} \sum_{L \subseteq S} (-1)^{|S|-|L|} v_{or}(x_{N \setminus L})$$

$$= -\sum_{L \subseteq N} \sum_{S:S \cap T \neq \emptyset, S \supseteq L} (-1)^{|S|-|L|} v_{or}(x_{N \setminus L})$$

$$= -\underbrace{v_{or}(x_\emptyset)}_{L=N} - \underbrace{v_{or}(x_T)}_{L=N \setminus T} \cdot \underbrace{\sum_{|S_2|=1}^{|T|} C_{|T|}^{|S_2|} (-1)^{|S_2|}}_{=-1}$$

$$- \sum_{L \cap T \neq \emptyset, L \neq N} v_{or}(x_{N \setminus L}) \cdot \underbrace{\sum_{S_1 \subseteq N \setminus T \setminus L} \sum_{|S_2|=|T \cap L|}^{|T|} C_{|T|-|T \cap L|}^{|S_2|-|T \cap L|} (-1)^{|S_1|+|S_2|}}_{=0}$$

$$- \sum_{L \cap T = \emptyset, L \neq N \setminus T} v_{or}(x_{N \setminus L}) \cdot \underbrace{\sum_{S_2 \subsetneq T} \sum_{S_1 \subseteq N \setminus T \setminus L} (-1)^{|S_1|+|S_2|}}_{=0}$$

$$= v_{or}(x_T) - v_{or}(x_\emptyset)$$

Therefore, $v_{or}(x_T) = \sum_{S \cap T \neq \emptyset, S \neq \emptyset} I_{or}(S) + v_{or}(x_\emptyset)$. In this way, we can derive that the output score $v(x_T)$ of the DNN on $x_T$ can be approximated by the sum of effects of AND-OR interactions.

$$v(x_T) = v_{and}(x_T) + v_{or}(x_T)$$

$$= \sum_{S \subseteq T} I_{and}(S) + \sum_{S \cap T \neq \emptyset, S \neq \emptyset} I_{or}(S) + v_{or}(x_\emptyset)$$

$$= \sum_{S \subseteq T, S \neq \emptyset} I_{and}(S) + v_{and}(x_\emptyset) + \sum_{S \cap T \neq \emptyset, S \neq \emptyset} I_{or}(S) + v_{or}(x_\emptyset)$$

$$= \sum_{S \subseteq T, S \neq \emptyset} I_{and}(S) + \sum_{S \cap T \neq \emptyset, S \neq \emptyset} I_{or}(S) + v(x_\emptyset)$$

# H  PROVING THAT THE SHAPLEY VALUE CAN BE EXPLAINED AS AND-OR INTERACTIONS

**Theorem 2.5.** *The Shapley value $\phi(i)$ of each input variable $i \in N$ can be explained as a re-allocation of AND-OR interactions,* i.e., $\phi(i) = \sum_{S \subseteq N, S \ni i} \frac{1}{|S|} I_{and}(S) + \sum_{S \subseteq N, S \ni i} \frac{1}{|S|} I_{or}(S)$.

• **proof:** The Shapley value of the input varibale $i$ is defined as follows. $\phi(i) = \mathbb{E}_{S \subseteq N \setminus \{i\}}[v(\boldsymbol{x}_{S \cup \{i\}}) - v(\boldsymbol{x}_S)]$. For simplicity, we use $v(S)$ to represent the advantage score $v(\boldsymbol{x}_S)$ on the masked board state $\boldsymbol{x}_S$.

According to Theorem 2.4, $\forall S \subseteq N, v(S) = v(\emptyset) + \sum_{T \subseteq S, T \neq \emptyset} I_{\text{and}}(T) + \sum_{T \cap S \neq \emptyset} I_{\text{or}}(T)$. Thus,

$$v(S \cup \{i\}) - v(S)$$

$$= \left[ v(\emptyset) + \sum_{T \subseteq (S \cup \{i\}), T \neq \emptyset} I_{\text{and}}(T) + \sum_{T \cap (S \cup \{i\}) \neq \emptyset} I_{\text{or}}(T) \right]$$

$$- \left[ v(\emptyset) + \sum_{T \subseteq S, T \neq \emptyset} I_{\text{and}}(T) + \sum_{T \cap S \neq \emptyset} I_{\text{or}}(T) \right]$$

$$= \left[ \sum_{T \subseteq (S \cup \{i\}), T \neq \emptyset} I_{\text{and}}(T) - \sum_{T \subseteq S, T \neq \emptyset} I_{\text{and}}(T) \right] + \left[ \sum_{T \cap (S \cup \{i\}) \neq \emptyset} I_{\text{or}}(T) - \sum_{T \cap S \neq \emptyset} I_{\text{or}}(T) \right]$$

$$= \underbrace{\sum_{T \subseteq S} I_{\text{and}}(T \cup \{i\})}_{\mathcal{A}} + \underbrace{\sum_{T \cap S = \emptyset} I_{\text{or}}(T \cup \{i\})}_{\mathcal{B}}$$

In this way, we can decompose the Shapley value $\phi(i)$ into two terms, *i.e.*, $\phi(i) = \mathbb{E}_{S \subseteq N \setminus \{i\}}[\mathcal{X} + \mathcal{Y}] = \mathbb{E}_{S \subseteq N \setminus \{i\}}[\mathcal{X}] + \mathbb{E}_{S \subseteq N \setminus \{i\}}[\mathcal{Y}]$. Next, we first analyze the sum of AND interactions $\mathbb{E}_{S \subseteq N \setminus \{i\}}[\mathcal{X}]$, and then analyze the sum of OR interactions $\mathbb{E}_{S \subseteq N \setminus \{i\}}[\mathcal{Y}]$.

$$\mathbb{E}_{S \subseteq N \setminus \{i\}}[\mathcal{X}]$$

$$= \mathbb{E}_{S \subseteq N \setminus \{i\}} \sum_{T \subseteq S} I_{\text{and}}(T \cup \{i\})$$

$$= \frac{1}{n} \sum_{m=0}^{n-1} \frac{1}{\binom{n-1}{m}} \sum_{\substack{S \subseteq N \setminus \{i\}, \, T \subseteq S \\ |S| = m}} \sum I_{\text{and}}(T \cup \{i\})$$

$$= \frac{1}{n} \sum_{T \subseteq N \setminus \{i\}} \sum_{m=0}^{n-1} \frac{1}{\binom{n-1}{m}} \sum_{\substack{S \supseteq T, \\ S \subseteq N \setminus \{i\}, \\ |S| = m}} I_{\text{and}}(T \cup \{i\})$$

$$= \frac{1}{n} \sum_{T \subseteq N \setminus \{i\}} \sum_{m=|T|}^{n-1} \frac{1}{\binom{n-1}{m}} \sum_{\substack{S \supseteq T, \\ S \subseteq N \setminus \{i\}, \\ |S| = m}} I_{\text{and}}(T \cup \{i\})$$

$$= \frac{1}{n} \sum_{T \subseteq N \setminus \{i\}} \sum_{m=|T|}^{n-1} \frac{1}{\binom{n-1}{m}} \binom{n-1-|T|}{m-|T|} I_{\text{and}}(T \cup \{i\})$$

$$= \frac{1}{n} \sum_{T \subseteq N \setminus \{i\}} \underbrace{\sum_{k=0}^{n-1-|T|} \frac{1}{\binom{n-1}{|T|+k}} \binom{n-1-|T|}{k}}_{\alpha_T} I_{\text{and}}(T \cup \{i\})$$

$$= \sum_{T \subseteq N \setminus \{i\}} \frac{1}{|T|+1} I_{\text{and}}(T \cup \{i\})$$

$$= \sum_{S \subseteq N, i \in S} \frac{1}{|S|} I_{\text{and}}(S) \quad // \text{ Let } S = T \cup \{i\}.$$

Then, for the sum of OR interactions, we have

$$\mathbb{E}_{S \subseteq N \setminus \{i\}}[\mathcal{B}]$$

$$= \mathbb{E}_{S \subseteq N \setminus \{i\}} \sum_{T \cap S \neq \emptyset} I_{\text{or}}(T \cup \{i\})$$

$$= \frac{1}{n} \sum_{m=0}^{n-1} \frac{1}{\binom{n-1}{m}} \sum_{\substack{S \subseteq N \setminus \{i\}, \, T \cap S \neq \emptyset \\ |S| = m}} \sum I_{\text{or}}(T \cup \{i\})$$

$$= \frac{1}{n} \sum_{T \subseteq N \setminus \{i\}} \sum_{m=0}^{n-1} \frac{1}{\binom{n-1}{m}} \sum_{\substack{S \cap T \neq \emptyset, \\ S \subseteq N \setminus \{i\}, \\ |S| = m}} I_{\text{or}}(T \cup \{i\})$$

$$= \frac{1}{n} \sum_{T \subseteq N \setminus \{i\}} \sum_{m=0}^{n-1} \frac{1}{\binom{n-1}{m}} \sum_{\substack{S \subseteq N \setminus \{i\} \setminus T, \\ |S| = m}} I_{\text{or}}(T \cup \{i\})$$

$$= \frac{1}{n} \sum_{T \subseteq N \setminus \{i\}} \sum_{m=0}^{n-1-|T|} \frac{1}{\binom{n-1}{m}} \sum_{\substack{S \subseteq N \setminus \{i\} \setminus T, \\ |S| = m}} I_{\text{or}}(T \cup \{i\}) \quad \text{// Since } S \subseteq N \setminus \{i\} \setminus T, |S| \leq n-1-|T|.$$

$$= \frac{1}{n} \sum_{T \subseteq N \setminus \{i\}} \sum_{m=0}^{n-1-|T|} \frac{1}{\binom{n-1}{m}} \binom{n-1-|T|}{m} I_{\text{or}}(T \cup \{i\})$$

$$= \frac{1}{n} \sum_{T \subseteq N \setminus \{i\}} \sum_{k=0}^{n-1-|T|} \frac{1}{\binom{n-1}{n-1-|T|-k}} \binom{n-1-|T|}{n-1-|T|-k} I_{\text{or}}(T \cup \{i\}) \quad \text{// Let } k = n-1-|T|-m.$$

$$= \frac{1}{n} \sum_{T \subseteq N \setminus \{i\}} \underbrace{\sum_{k=0}^{n-1-|T|} \frac{1}{\binom{n-1}{|T|+k}} \binom{n-1-|T|}{k}}_{\alpha_T} I_{\text{or}}(T \cup \{i\})$$

$$= \frac{1}{n} \sum_{T \subseteq N \setminus \{i\}} \frac{n}{|T|+1} I_{\text{or}}(T \cup \{i\})$$

$$= \sum_{T \subseteq N \setminus \{i\}} \frac{1}{|T|+1} I_{\text{or}}(T \cup \{i\})$$

$$= \sum_{S \subseteq N, i \in S} \frac{1}{|S|} I_{\text{or}}(S) \quad \text{// Let } S = T \cup \{i\}.$$

Therefore, $\phi(i) = \sum_{S \subseteq N \setminus \{i\}}[\mathcal{X}] + \sum_{S \subseteq N \setminus \{i\}}[\mathcal{Y}] = \sum_{S \subseteq N, S \ni i} \frac{1}{|S|} I_{\text{and}}(S) + \sum_{S \subseteq N, S \ni i} \frac{1}{|S|} I_{\text{or}}(S)$.

## I   PROVING THAT EXCLUSIVELY USING AND INTERACTIONS TO EXPLAIN THE OR RELATIONSHIP WILL SIGNIFICANTLY COMPLICATE THE EXPLANATION

**Theorem 2.2.** *Given an input sample* $x = [x_1, x_2, ..., x_n]^\top$*, where each input variable* $x_{k_j} \in \{0, 1\}$ *is binary to represent the presence or absence state of the variable. Let the target function* $v(x)$ *encode a m-order OR relationship, i.e.,* $v(x) = x_{k_1} \vee x_{k_2} \vee ... \vee x_{k_m}$*. Then, the function* $v(x)$ *would activate* $2^m - 1$ *non-zero AND interactions, i.e. for all non-empty subset,* $\emptyset \neq S \subseteq \{x_{k_1}, x_{k_2}, ..., x_{k_m}\}, I_{and}(S|x) = (-1)^{|S|-1} \neq 0$.

**On one hand, incorrect baseline values generate an exponential number of incorrect interactions.** Let us construct a toy function $v(\boldsymbol{x}) = w_S \prod_{j \in S}(x_j - \alpha_j)$ for analysis. In the constructed toy function, we can consider $\alpha_j$ as the ground-truth baseline value for the input variable $x_j$. We use

$U(S) = \sum_{T \subseteq S} (-1)^{|S|-|T|} v(\boldsymbol{x}_T)$ to compute the effect of the interaction $S$. Based on the ground-truth baseline values $\{\alpha_j\}_{j \in S}$, there is only one interaction primitive with non-zero effect $U(S)$. In comparison, if we use $m$ incorrect baseline values $\{\beta_j\}$ to compute the effect of the interaction primitives, subject to $\sum_{j \in S} \mathbb{1}_{\beta_j \neq \alpha_j} = m$, then the function will be explained to contain at most $2^m$ salient primitives. In other words, **incorrect baseline values will generate an exponential number of incorrect interactions, which complicate the explanation.**

● **proof:** Without loss of generality, we consider an input sample $\boldsymbol{x}$, with $\forall j \in S, x_j \neq \alpha_j$. Based on the ground-truth baseline value $\{\alpha_j\}$.

(1) First, we have $v(\boldsymbol{x}_S) = w_S \prod_{j \in S}(x_j - \alpha_j) \neq 0$.

(2) Second, we have $\forall T \subsetneq S, v(\boldsymbol{x}_T) = w_S \prod_{j \in T}(x_j - \alpha_j) \prod_{k \in S \setminus T}(\alpha_k - \alpha_k) = 0$. Then, we have $\forall T \subsetneq S$, we have $U(T) = \sum_{L \subseteq T}(-1)^{|T|-|L|} v(\boldsymbol{x}_L) = \sum_{L \subseteq T} 0 = 0$, and we have $U(S) = \sum_{T \subseteq S}(-1)^{|S|-|T|} v(\boldsymbol{x}_T) = v(\boldsymbol{x}_S) \neq 0$.

(3) Third, $\forall T \neq S$, let $T = L \cup M$, where $L \subseteq S$ and $M \cap S = \emptyset$. Then, we have

$$
\begin{aligned}
U(T) &= \sum_{S' \subseteq T} (-1)^{|T|-|S'|} v(\boldsymbol{x}_{S'}) \\
&= \sum_{\substack{L' \subseteq L \\ L' \neq \emptyset}} (-1)^{|T|-|L'|} v(\boldsymbol{x}_{L'}) + \sum_{\substack{M' \subseteq M \\ M' \neq \emptyset}} (-1)^{|T|-|M'|} \underbrace{v(\boldsymbol{x}_{M'})}_{=v(\boldsymbol{x}_\emptyset)=0} \\
&\quad + \sum_{\substack{L' \subseteq L, M' \subseteq M \\ L' \neq \emptyset, M' \neq \emptyset}} (-1)^{|S|-|L'|-|M'|} \underbrace{v(\boldsymbol{x}_{L' \cup M'})}_{=v(L')} + (-1)^{|T|} \underbrace{v(\boldsymbol{x}_\emptyset)}_{=0} \\
&= \sum_{\substack{L' \subseteq L \\ L' \neq \emptyset}} (-1)^{|T|-|L'|} v(\boldsymbol{x}_{L'}) + \sum_{\substack{L' \subseteq L, M' \subseteq M \\ L' \neq \emptyset, M' \neq \emptyset}} (-1)^{|S|-|L'|-|M'|} v(\boldsymbol{x}_{L'}) \\
&= (-1)^{|T|-|S|} v(\boldsymbol{x}_S) + \sum_{\substack{M' \subseteq M \\ M' \neq \emptyset}} (-1)^{|T|-|S|-|M'|} v(\boldsymbol{x}_S) \quad \% \; v(\boldsymbol{x}_{L'}) \neq 0 \text{ only if } L' = S \\
&= \sum_{M' \subseteq M} (-1)^{|T|-|S|-|M'|} v(\boldsymbol{x}_S) = 0
\end{aligned}
$$

Therefore, there is only one interaction primitive with non-zero effect $U(S)$.

In comparison, if we use $m$ incorrect baseline values $\{\beta_j\}$ to compute the effect of the interaction primitives, where $\sum_{j \in S} \mathbb{1}_{\beta_j \neq \alpha_j} = m$, then the function will be explained to contain at most $2^m$ interaction primitives with salient effects. For the simplicity of notations, let $S = \{1, 2, ..., m\}$, and $\beta_1 = \alpha_1 + \epsilon_1, ..., \beta_m = \alpha_m + \epsilon_m$, where $\epsilon_1, ..., \epsilon_m \neq 0$. Let $T = \{1, 2, \ldots, m\}$.

In the case of using $m$ incorrect baseline values,
(1) First, we have $v(\boldsymbol{x}_S) \neq 0$
(2) Second, we have $\forall T \subsetneq S, |T| < n - m, v(\boldsymbol{x}_T) = w_S \prod_{j \in T}(x_j - \alpha_j) \prod_{l \in S \setminus T}(\beta_l - \alpha_l)$. Because $|S| - |T| > m$, there is at least one variable with ground-truth baseline value in $S \setminus T$. Therefore, $v(\boldsymbol{x}_T) = 0$. Furthermore, $U(T) = \sum_{L \subseteq T}(-1)^{|T|-|L|} v(\boldsymbol{x}_L) = 0$
(3) $\forall T \subsetneq S, |T| = k \geq n - m, v(\boldsymbol{x}_T) = w_S \prod_{j \in T}(x_j - \alpha_j) \prod_{l \in S \setminus T}(\beta_l - \alpha_l)$. If $S \setminus S' \subseteq T$, then $S \setminus T \subseteq S'$ and $v(\boldsymbol{x}_T) \neq 0$. Otherwise, $v(\boldsymbol{x}_T) = 0$. Then,

$$
\begin{aligned}
U(T) &= \sum_{L \subseteq T}(-1)^{|T|-|L|} v(\boldsymbol{x}_L) \\
&= \sum_{L \subseteq T, |L| < n-m}(-1)^{|T|-|L|} v(\boldsymbol{x}_L) + \sum_{L \subseteq T, L \geq n-m}(-1)^{|T|-|L|} v(\boldsymbol{x}_L) \\
&= 0 + \sum_{L \subseteq T, L \geq n-m, L \supseteq S \setminus S'}(-1)^{|T|-|L|} v(\boldsymbol{x}_L) + \sum_{L \subseteq T, L \geq n-m, L \not\supseteq S \setminus S'}(-1)^{|T|-|L|} v(\boldsymbol{x}_L) \\
&= \sum_{L \subseteq T, L \geq n-m, L \supseteq S \setminus S'}(-1)^{|T|-|L|} v(\boldsymbol{x}_L)
\end{aligned}
$$

If the above $U(T)) = 0$, it indicates that $S \setminus S' \nsubseteq T$. In this case, there is no subset $L \subseteq T$ s.t. $S \setminus S' \subseteq L$. In other words, only if $S \setminus S' \subseteq T, U_T \neq 0$. In this way, a total of $\binom{m}{k-(|S|-m)}$ causal patterns of the $k$-th order emerge, where the order $k$ of a causal pattern means that this causal pattern $T$ contains $k = |T|$ variables. There are totally $\sum_{k=|S|-m}^{n} \binom{m}{k-(|S|-m)} = 2^m$ causal patterns in $x$.

For example, if the input $x$ is given as follows,

$$x_i = \begin{cases} \alpha_i + 2\epsilon_i, & i \in S' = \{1, \ldots, m\} \\ \alpha_i + \epsilon_i, & i \in S \setminus S' = \{m+1, \ldots, n\} \end{cases}$$

where $\epsilon_i \neq 0$ are arbitrary non-zero scalars. In this case, we have $\forall T \subseteq S', U(T \cup \{m+1, ..., n\}) = \epsilon_1 \epsilon_2 ... \epsilon_n \neq 0$. Besides, if $\{m+1, ..., n\} \nsubseteq T$, we have $U(T) = 0$.

In this way, there are totally $2^m$ causal patterns in $x$.

This explains the motivation of extending AND interactions to OR interactions. It is because the OR interaction can be considered as a specific AND interaction, where the presence of a stone is taken as the masked state, and the removing of a stone is considered as the unmasked state.

**On the other hand, without defining the OR interaction, we have to use $2^m$ AND interactions to represent a single OR relationship between input variables.** This is a more direct explanation for the utility of using OR interactions. Let us focus on a toy function $v(x) = \vee_{j \in S} \mathrm{unmask}(x_j)$ that represents an single OR relationship between input variables in $S$. In this case, if we explain $v(x)$ by exclusively using AND interactions, then we will obtain a total of $2^m - 1$ AND interactions, subject to $m = |S|$. For each subset $T \subseteq S, T \neq \emptyset$, we will get a non-zero interaction effect $I(T) = \sum_{k=0}^{|T|-1} (-1)^k \cdot C_{|T|}^k = (-1)^{|T|-1}$.

Therefore, to faithfully represent the OR relationship encoded by the KataGo model, we must extend AND interactions to OR interactions. Otherwise, the explanation will be significantly complicated.

## J    THE REASON WHY THE SATURATION PROBLEM CAUSES HIGH-ORDER INTERACTIONS

Let $e_k \overset{\text{def}}{=} \mathbb{E}_x \mathbb{E}_{T \subseteq N : \Delta n = k} \log(\frac{p_{\text{white}}(x_T)}{1 - p_{\text{white}}(x_T)})$ denote the average advantage score over all masked states $x_T$ with the same unbalance level $k$. Let $g \in R$ and $h \in R$ denote the first derivative and second derivative of the curve of $e_k \overset{\text{def}}{=} \mathbb{E}_x \mathbb{E}_{T \subseteq N : \Delta n = k} \log(\frac{p_{\text{white}}(x_T)}{1 - p_{\text{white}}(x_T)})$ w.r.t. the $k$ value ($k \in \{-\frac{n}{2}, -\frac{n}{2} + 1, ..., \frac{n}{2}\}$). Then, we can roughly consider that $e_k = e_0 + g \cdot k + \frac{h}{2} \cdot k^2$.

Let us consider an interaction $S$ between $m$ stones, including $m_{\text{white}}$ white stones and $m_{\text{black}}$ black stones. The unbalance level of the masked board state $x_S$ is $\Delta n = m_{\text{white}} - m_{\text{black}} = k^*$. If we only use AND interactions to explain the output of the value network, then we obtain the following equation.

$$v_m \overset{\text{def}}{=} \mathbb{E}_{T \subseteq S : |T| = m}[v(x_T)] \approx e_{k^*}$$

$$v_{m'} \overset{\text{def}}{=} \mathbb{E}_{T \subseteq S : |T| = m'}[v(x_T)]$$

$$\approx \frac{e_{k^* - ((m-m'))}}{\binom{m_{\text{white}}}{m-m'}} + \frac{e_{k^* - ((m-m'-1))}}{\binom{m_{\text{white}}}{m-m'-1}} + \ldots + \frac{e_{k^* + ((m-m'-1))}}{\binom{m_{\text{black}}}{m-m'-1}} + \frac{e_{k^* + ((m-m'))}}{\binom{m_{\text{black}}}{m-m'}} \quad (8)$$

$$v_0 \overset{\text{def}}{=} \mathbb{E}_{T \subseteq S : |T| = 0}[v(x_T)] \approx e_0$$

Note that $v_m, v_{m'}$ and $v_0$ are non-linear functions. The function $v_{m'}$ can be rewritten by following Taylor series expansion at the baseline point $m' = 0$ as follows.

$$v_{m'} = \mathbb{E}_{T \subseteq S : |T| = m'}[v(x_T)] = v_0 + g_v \cdot m' + \frac{h_v}{2} \cdot m'^2 \quad (9)$$

where $g_v \in R$ and $h_v \in R$ denote the first derivative and second derivative of the curve of $v_{m'}$ w.r.t. the $m'$ value. In this way, the effect $I(S)$ of the interaction $S$ can be reformulated as follows.

$$
\begin{aligned}
I(S) &= \sum_{T \subseteq S} (-1)^{|S|-|T|} v(\boldsymbol{x}_T) \\
&\approx \binom{m}{0} v_m - \binom{m}{1} \cdot v_{m-1} + \binom{m}{2} \cdot v_{m-2} - \binom{m}{3} \cdot v_{m-3} + \binom{m}{4} \cdot v_{m-4} - ...
\end{aligned}
\tag{10}
$$

According to Equation (9), each component $v_{m'}$ of $I(S)$ consists of a term $\frac{h_v}{2} \cdot m'^2$. However, the term $\frac{h_v}{2} \cdot m'^2$ contained in $v_{m'}$ cannot cancel out with each other. Therefore, the interaction effect $I(S)$ will increase with the order of the primitive $S$.

## K    THEOREMS AND PROPERTIES OF THE ATTRIBUTION METHOD IN EQUATION (6).

The coalition attribution satisfies the following desirable properties.

• **Symmetry property:** If the input variable $i \in N$ and the input variable $j \in N$ cooperate with other input variables in $S \subseteq N \setminus \{i,j\}$ in the same way, *i.e.* $\forall S \subseteq N \setminus \{i,j\}, v(S \cup \{i\}) = v(S \cup \{j\})$, then the coalition formed by $S \cup \{i\}$ and the coalition formed by $S \cup \{j\}$ have the same attribution, *i.e.*, $\forall S \subseteq N \setminus \{i,j\}, \varphi(S \cup \{i\}) = \varphi(S \cup \{j\})$.

• **Additivity property:** If the output score of the model $v$ can be represented as the sum of the output score of the model $v_1$ and the output score of the model $v_2$, *i.e.* $\forall S \subseteq N, \ v(S) = v_1(S) + v_2(S)$, then the attribution of any coalition $S$ on the model $v$ can also be represented as the sum of the attribution of $S$ on the model $v_1$ and that on the model $v_2$, *i.e.* $\forall S \subseteq N, \ \varphi_v(S) = \varphi_{v_1}(S) + \varphi_{v_2}(S)$.

• **Dummy property:** If a coalition $S$ is a dummy coalition, *i.e.* $\forall i \in S, \forall T \subseteq N \setminus \{i\}, v(T \cup \{i\}) = v(T)$, then the coalition $S$ has no attribution on the model output, *i.e.* $\varphi(S) = 0$.

• **Efficiency property:** For any coalition $S$, the model output can be decomposed into the attribution of the coalition $S$ and the attribution of each input variable in $N \setminus S$ and the utilities of the interactions covering partial variables in $S$, *i.e.*, $\forall S \subseteq N, v(N) - v(\emptyset) = \varphi(S) + \sum_{i \in N \setminus S} \varphi(i) + \sum_{T \subseteq N, T \cap S \neq \emptyset, T \cap S \neq S} \frac{|T \cap S|}{|T|} [I_{and}(T) + I_{or}(T)]$

And we try to use Corollary K.1 and Equation (7) to explain the conflict between the Shapley value of input variables and the attribution of the coalition as follows.

**Corollary K.1.** *If* $\forall T \subseteq N, T \ni i, T \not\supseteq S, I_{and}(T) = I_{or}(T) = 0$, *then* $\phi(i) = \frac{1}{|S|} \varphi(S)$

Corollary K.1 shows that if a set $S$ of input variables is always memorized by the DNN as a coalition, and the DNN does not encode any interactions between a set $T$ of input variables, where $T$ only contains partial variables in $S$, *i.e.*, $\forall T \subseteq N, T \cap S \neq S, T \cap S \neq \emptyset, I_{and}(T) = I_{or}(T) = 0$, then the attribution $\varphi(S)$ of the coalition $S$ can be fully determined by the sum of the Shapley value $\phi(i)$ of all input variables in $S$. Otherwise, if the DNN encodes interactions between a set $T$ of input variables, where $T$ contains just partial but not all variables in $S$, then Equation (7) shows the conflict between individual variables' attributions and the coalition $S$'s attribution come from interactions containing just partial but not all variables in $S$.

## L    EXPERIMENTAL DETAILS

### L.1    THE COMPUTER RESOURCES

Our experimental are conducted on a server, which is equipped with a CPU that has 4 cores and 16 threads, 126GB of RAM, and an SSD with 960GB capacity. The server also includes two NVIDIA GeForce RTX Titan GPUs.

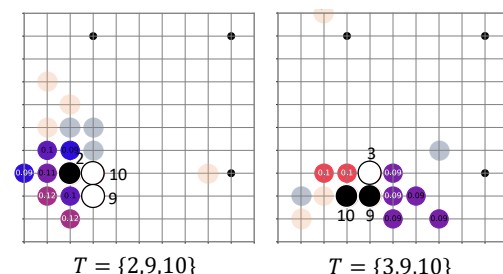

$$T = \{2,9,10\} \qquad T = \{3,9,10\}$$

Figure 8: Attribution values of the inter-action context of the target coalition.

## L.2 SETTINGS FOR THE GENERATION OF ONE BOARD CONFIGURATION

We use pre-trained networks published on https://github.com/lightvector/KataGo. We set the board size as 19x19, by letting the KataGo play games against itself, *i.e.*, letting the KataGo take turns to play the move of black stones and play the move of white stones, we can generate a board state.

## L.3 SETTINGS FOR THE EXTRACTION OF INTERACTIONS IN SECTION 2.2

The learning rate for the learnable vector $\boldsymbol{p}$, $\boldsymbol{q}$ exponentially decays from 1e-6 to 1e-7. In particular, each element $a_k$ in the vector $\boldsymbol{a}$ has different initial learning rates. Specifically, the learning rate of $a_k$ decayed from $\frac{1}{\binom{|k|}{\frac{n}{2}}} \cdot 1e-6$ to $\frac{1}{\binom{|k|}{\frac{n}{2}}} \cdot 1e-7$.

The threshold $\tau$ is a small scalar to bound unavoidable noises $\boldsymbol{q}$ in the network output, which is set to be $\tau = 0.38$ in experiments, which is set to be 0.01 time of the average strength of the top-1% most salient interaction. Specifically, we compute all AND interactions $\{I_{\text{and}}(S_1), I_{\text{and}}(S_2), ..., I_{\text{and}}(S_{2^n})\}$ by setting $v_{\text{and}}(\boldsymbol{x}_T)$ as $v(\boldsymbol{x}_T)$, and compute all OR interactions $\{I_{\text{or}}(S_1), I_{\text{or}}(S_2), ..., I_{\text{or}}(S_{2^n})\}$ by setting $v_{\text{or}}(\boldsymbol{x}_T)$ in Equation (4) as $v(\boldsymbol{x}_T)$. Then, all AND interactions $\{I_{\text{and}}(S_1), I_{\text{and}}(S_2), ..., I_{\text{and}}(S_{2^n})\}$ and all OR interactions $\{I_{\text{or}}(S_1), I_{\text{or}}(S_2), ..., I_{\text{or}}(S_{2^n})\}$ are arranged in descending order of their interaction strength.

## L.4 COMPUTING THE ATTRIBUTION OF THE INTERACTION CONTEXT.

The attribution of the stone in the interaction context can be computed as:

$$\text{attribution}(x_i) = \sum_{S \ni i} \frac{|I(S)|}{|S|} \qquad (11)$$

We compute the attribution of contextual stones to a target coalition $S$, and we visualizes attribution values of contextual stones in Figure 8.

# M MORE EXPERIMENTAL RESULTS

## M.1 TRANSFERABILITY OF INTERACTIONS BETWEEN STONES THROUGH DIFFERENT BOARDS.

We conduct experiments to show the transferability of interactions between stones through different boards. As Figure 7 shows, the explained stones in game board states 1, 2, 3 are the same, but the contextual stones are different. We computed interactions between the same set of stones on different boards (with different contextual stones). Figure 7 shows that the same interaction exhibited similar effects on different boards. For example, for Game 1, $\varphi(\{1,2,9\}) = 1.58$, for Game 2, $\varphi(\{1,2,9\}) = 1.49$, and for Game 3, $\varphi(\{1,2,9\} = 1.50$. It means that the shape patterns found on one board can be transferred to another board.

## M.2 MORE SHAPE PATTERNS EXTRACTED FROM THE VALUE NETWORK FOR THE GAME OF GO.

We show more shape patterns extracted from the value network for the game of Go.

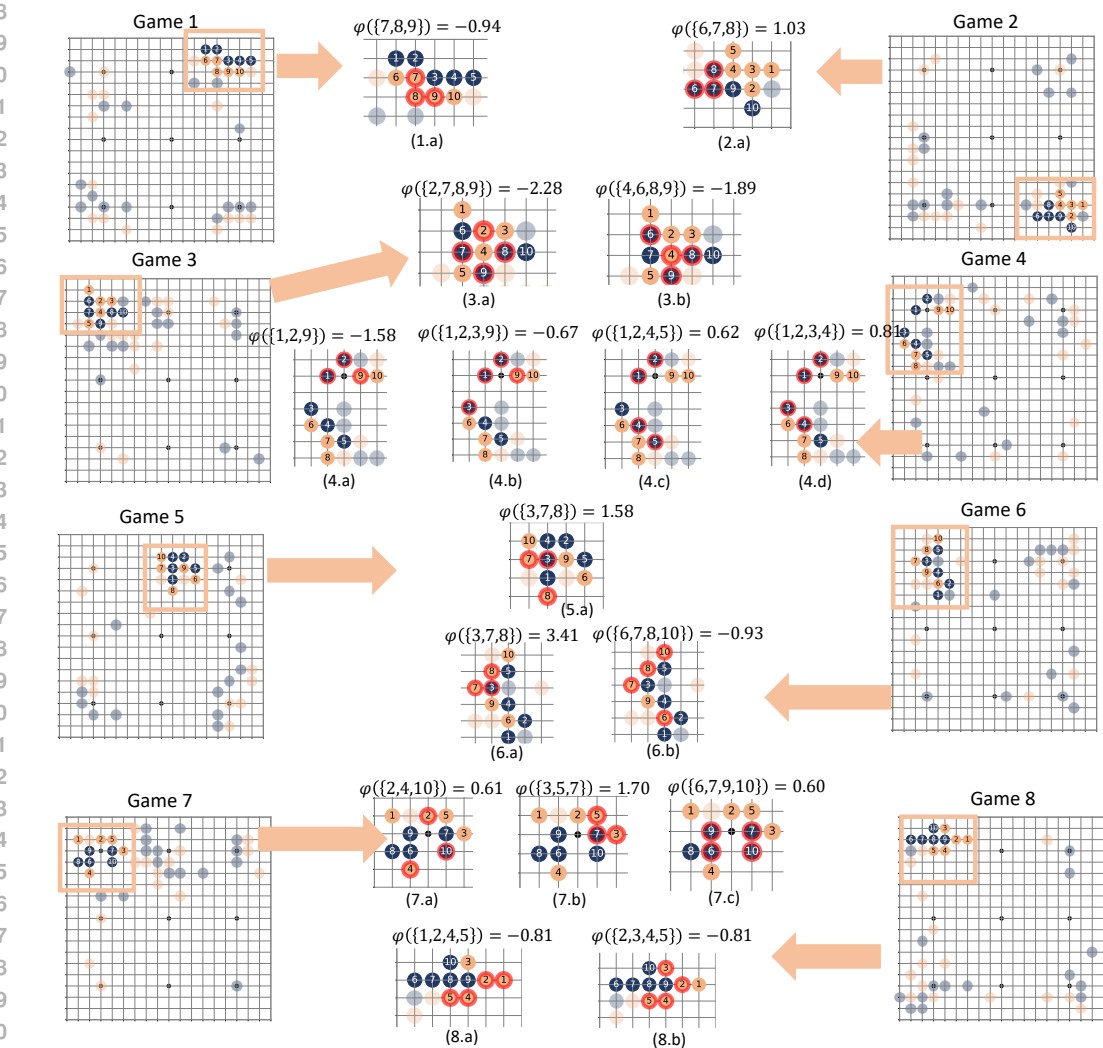

Figure 9: More experimental results for the estimated attributions of different coalitions (shape patterns). Stones in the coalition are high-lighted by red circles.

**For Game 1** in Figure 9 (1.a), Go players are confused about why the coalition $\{7, 8, 9\}$ is advantageous for black stones.

**For Game 2** in Figure 9 (2.a), Go players cannot figure out why the coalition $\{6, 7, 8\}$ is advantageous for white stones.

**For Game 3** in Figure 9 (3.a-3.b), Go players consider that the black stones $x_6$, $x_7$ are caught, and the white stones are in advantage. However, the value network think that the coalition $\{2, 7, 8, 9\}$ and the coalition $\{4, 6, 8, 9\}$ are advantageous for black stones. Go players are confused about that.

**For Game 4** in Figure 9 (4.a-4.d), $\varphi(\{1, 2, 9\}) < \varphi(\{1, 2, 3, 9\})$, which means that the black stone $x_3$ is a low-value move, Go players consider that the stone $x_3$ a valuable move.

**For Game 5** in Figure 9 (5.a), $\varphi(\{3, 7, 8\}) = 1.58$ means that the white stones are in advantage. Furthermore, this shape pattern is considered a classic shape among Go players, aligning with their strategic understanding of the game.

**For Game 6** in Figure 9 (6.a-6.b), $\varphi(\{6, 7, 8, 10\}) = -0.93$, it is confused for Go players that the black stones are in advantage in the shape pattern $\{6, 7, 8, 10\}$. While $\varphi(\{3, 7, 8\}) = 3.41$, the shape

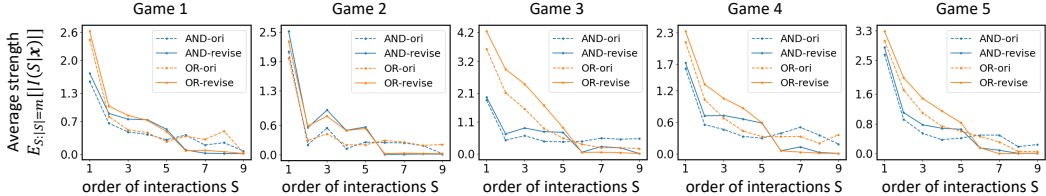

Figure 10: Average strength of effects for interactions of different orders. For different gobang games, our revised method extracts weaker high-order interactions than the original method.

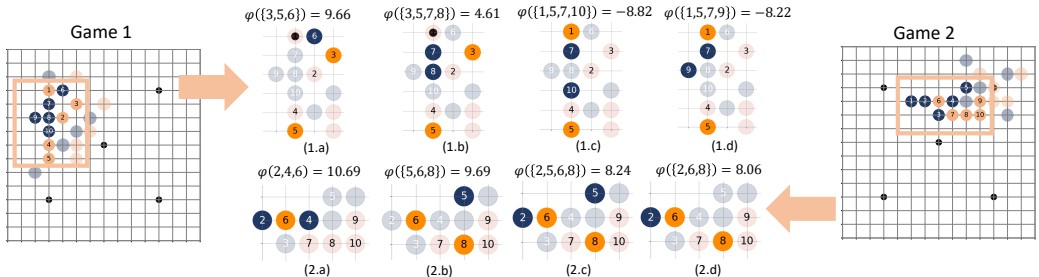

Figure 11: Estimated attributions of different coalitions (shape patterns) selected from two gobang game states.

pattern $\{3, 7, 8\}$ is a classic shape among Go players, and the attribution score align with Go players' strategic understanding of the game.

**For Game 7** in Figure 9 (7.a-7.c), $\varphi(\{2, 4, 10\}) = 0.61$ and $\varphi(\{3, 5, 7\}) = 1.70$, which means that white stones are in advantage in the shape pattern $\{2, 4, 10\}$ and $\{3, 5, 7\}$. Meanwhile, Go players consider that $\{2, 4, 10\}$ and $\{3, 5, 7\}$ are classic shape patterns, and the attribution scores of these two coalition align with their understanding of the Go game. However, the shape pattern $\{6, 7, 9, 10\}$ consists of four black stones, while $\varphi(\{6, 7, 9, 10\}) = 0.60$ means that white stones are in advantage in the pattern $\{6, 7, 9, 10\}$, which contracts with the understanding of Go player.

**For Game 8** in Figure 9 (8.a-8.b), $\varphi(\{1, 2, 4, 5\}) = -0.81$ and $\varphi(\{2, 3, 4, 5\}) = -0.81$, which means that black stones are in advantage in these two shape patterns. However, these two shape patterns $\{1, 2, 4, 5\}$ and $\{2, 3, 4, 5\}$ are both composed of four white stones. It makes it difficult for Go palyers to understand.

### M.3 EXTENSION OF OUR METHOD TO EXPLAIN GOBANG GAME

We apply our method to another application, i.e., the Gobang game. Specifically, we analyze the value network in the open-source Gobang project, Katagomo. The Katagomo model is designed upon the KataGo project, and it also has a value network. We use the Katagomo model to generate a Gobang board with 20 stones. Then, we use our method to analyze shape patterns encoded by Katagomo's value network. Furthermore, we compute the attribution $\varphi(T)$ of each shape pattern encoded by the katagomo's value network. Figure 10 compares the strength of interactions of different orders. Given different Gobang game board states, our method in Equation (5) extracts weaker high-order interactions than the original method in Equation (4). Figure 11 visualizes the common coalitions selected from two gobang game states.

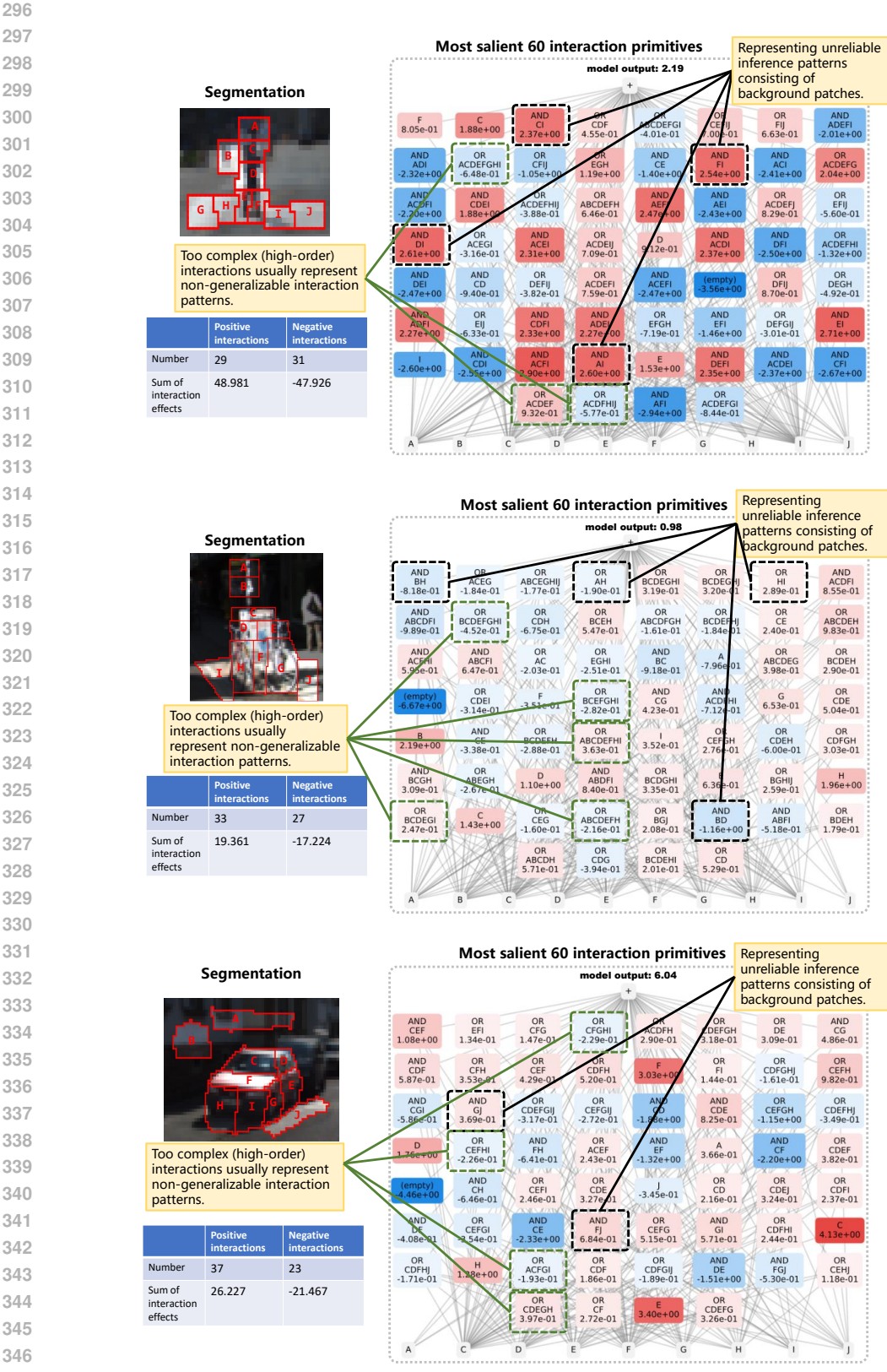

Figure 12: The interactions extracted from a DNN for pedestrian detection.

## M.4 Extension of our method to debug representation flaws hidden in a DNN

The second application of our method is to debug representation flaws hidden in a DNN. Besides letting people learn new interactions/concepts encoded by the DNNs, another typical utility of explaining interactions in a DNN is to debug representation flaws hidden in a DNN.

Figure 12 shows the interactions extracted from a DNN for pedestrian detection. Given an input image, we manually label image regions with salient attributions as input variables, and compute interactions between image regions. The visualization of the interactions enables people to check the correctness of interactions encoded by the DNN manually. Let us consider the explanation on the first input image as an example. We can analyze the representation quality of the DNN from the following three perspectives. (1) The interactions $I_{\text{and}}(S = \{C, I\})$, $I_{\text{and}}(S = \{D, I\})$, $I_{\text{and}}(S = \{F, I\})$ and $I_{\text{and}}(S = \{A, I\})$ between pedestrian patches and background patches may represent unreliable inference patterns. (2) High-order interactions, $e.g.$, $I_{\text{or}}(S = \{A, C, D, E, F, G, H, I\})$ and $I_{\text{or}}(S = \{A, C, D, F, H, I, J\})$, usually represent too complex inference patterns. Complex interactions usually have lower generalization power than simple interactions. (3) There are 29 positive interactions and 31 negative interactions extracted from an input image. The offsetting of positive and negative interactions is another problem. Adversarially robust neural networks usually encode more positive interactions and fewer negative interactions than normal neural networks.

In addition, the problematic interactions ($e.g.$, interactions on background patches) reflect representation flaws of a DNN, because it is found by Li & Zhang (2023) that salient interactions are usually transferable across different samples. In other words, problematic interactions may affect the inference of a large number of samples.

