# OpenReview forum: "Disentangling the QiGan Encoded by a DNN towards the Go Game"
_ICLR.cc/2025/Conference — ICLR 2025 Conference Withdrawn Submission_

### Official Review · Reviewer_Df8Q · 2024-11-02

**Soundness:** 2
**Presentation:** 1
**Contribution:** 3
**Rating:** 3
**Confidence:** 4

**Summary:**

This paper introduces a method to disentangle the knowledge encoded in DNNs, specifically applied to AI models for the game of Go.
The author introduces the concept of "QiGan" which is analyzed to reveal the AI’s decision-making process. The authors propose an approach for extracting these patterns to enhance human understanding of AI strategies beyond traditional human knowledge.

**Strengths:**

1. Effort is shown in evaluating games by professional Go players.
2. Visual aids are effectively used to present the complex Go game.

**Weaknesses:**

This paper is not well-written and is plagued with flawed expressions and grammatical errors. It is also not well-organized enough to convey the logical progression of the idea.

1. The concept of "QiGan", a new concept introduced by the author, is not explained in detail nor properly cited. The paper lacks a clear exposition that distinguishes if "QiGan" refers to intuition, "knowledge" (line 052) , or "understanding" (line 423) in the context of the Go game.

2. The notion of "AND" and "OR" interactions, which have been frequently used and employed as the backbone of the explanation in the manuscript, is not properly defined nor referred to.

3. The experiments lack quantitative rigor and fail to provide a clear metric for validating the claims made about the QiGan disentanglement. The paper would benefit significantly from a more robust experimental framework to substantiate its claims.

**Questions:**

1. While the visual presentation of Go game boards is helpful, could the author clarify how these visuals directly relate to the "QiGan" concepts being discussed?

---

### Official Review · Reviewer_gs4T · 2024-11-04

**Soundness:** 2
**Presentation:** 2
**Contribution:** 2
**Rating:** 5
**Confidence:** 3

**Summary:**

In this work, the authors try to analyze pre-trained Go models. They rely on the AND interactions [1] framework, which they extend by incorporating OR relations. These relations describe the existence or absence of a set of Go stones.
Further, a dive into pre-trained Go models revealed biases in those models in cases where a) white stones are far more or less than black stones and b) shapes consisting of a large group of stones. They also propose a methodology to alleviate a) and collaborate with Go players to recognize novel/unlikely Go states provided by the pre-trained models.


[1] Ren, Qihan, et al. "Where We Have Arrived in Proving the Emergence of Sparse Interaction Primitives in DNNs." The Twelfth International Conference on Learning Representations.

**Strengths:**

Some strengths of this work include:
- Developing methodologies that offer insights into pre-trained Go models.
- Investigating the use of pre-trained Go models to generate novel strategies for human Go players.
- The authors did not restrict their analysis to Go models; they also aimed to apply their methodology to the game of Go Bang and to object detection models.

**Weaknesses:**

Some weaknesses of this work are as follows:

- The paper's readability could be significantly improved. The reviewer believes that the authors should provide additional details or citations for topics that have been introduced in prior research, such as AND interactions.

- The main paper lacks a dedicated related work section. The reviewer suggests that this section be included in the main paper.

- Section 2.5 is missing a quantitative analysis. The authors offer a few examples to demonstrate whether their approach can detect meaningful Go states. However, the reviewer believes that this analysis should be enhanced by including an evaluation protocol. For instance, how many Go players are assessing these states? What qualifications do the players need to be considered professionals? A substantial number of examples should be extracted from actual Go games. Additionally, it is crucial to clarify what information is provided to the human players: are they given only the Go state or both the Go state and the v(x) values? This clarity is vital for enhancing the scientific quality and reproducibility of the work.

- Figure 2 could be improved. The reviewer recommends combining the two sub-figures into one to clearly illustrate the increase in the sparsity of interactions when both AND and OR interactions are considered.

**Questions:**

- Algorithm 1 involves a manual annotation process. Who are the annotators? Are they the authors or professional human players?
- Additionally, for Algorithm 1, how is the initial board state chosen for your experiments?

---

### Official Review · Reviewer_9Hui · 2024-11-04

**Soundness:** 2
**Presentation:** 1
**Contribution:** 1
**Rating:** 3
**Confidence:** 3

**Summary:**

The authors attempt to analyze the solutions learned by KataGo, a DNN-based model, to solve the game of GO. To do this they develop a mathematical framework to analyze the model and its solutions in Go. They then offer some analysis of Go behavior learned by the network and qualitatively compare it to humans.

**Strengths:**

The authors attempt a very challenging problem: using a DNN that performs better than humans on Go to derive new strategies for humans to use to solve the game. This is an awesome direction, which has been attempted previously albeit sparingly, and which — if made systematic and generalizable — could prove to be an enormous contribution. Indeed, the authors mention that they generalize their method to Gobang, which is a different game played on the same board as Go. I encourage the authors to continue this work as it could be very impactful.

**Weaknesses:**

I found this paper to be inscrutable. The goal is commendable, but a paper about interpretability should not be so challenging to understand. I would recommend moving most of the theorems to the SI. Simplify your explanation of the method into an intuition, which you back up with references to specific theorems and proofs. Top-line the human/model study instead of the method that you used to derive those insights. Use the figures to highlight strategies taken from the model. In other words, follow the template of "Acquisition of Chess Knowledge in AlphaZero." I honestly don't know how to read this paper because it is so dense and so challenging to parse in its current form.

Relatedly, the writing needs a lot of work. Beyond just the organization of the paper, and top-lining the main points, lots of the language is challenging to parse. For example, the italicized sentence in the intro is very challenging to parse. This should be written as simply as possible to make the paper more impactful.

This is a purely qualitative paper, which adds to the challenge of interpreting it. There needs to be some hypothesis testing in order to show that the methods and insights are indeed significant.

**Questions:**

See the above. The paper is such a challenge to understand I could not even come up with clear questions about the content.

---

### Official Review · Reviewer_EFhJ · 2024-11-05

**Soundness:** 4
**Presentation:** 4
**Contribution:** 3
**Rating:** 6
**Confidence:** 2

**Summary:**

1) This paper presents a novel method to extract the knowledge learned by a DNN value network for the game of Go. The proposed method can extract learned shape patterns in KataGo an OSS model trained to play Go. The extracted shape patterns align with the intuition of human players in most cases but also lead to new intuitions.

2) The authors make several key contributions to ensure the extracted shape patterns are not spurious.
 a) Previously it was shown that Go board state value predictions can be formulated using AND interactions between the stones but it is not sufficient so the authors introduce OR interactions.
 b) They make sure the universal matching property is satisfied i.e no matter how you mask the board state an accurate value prediction can be made using the discovered shape patterns.
c) Authors made sure the extracted interactions are higher order such that they are transferrable between board states.

**Strengths:**

Paper is well-written with clear to understand pre-requisites explained before getting into the core contributions.

**Weaknesses:**

Authors manually annotate common coalitions which requires a high degree of skill in playing Go. It'd be nice if this step could be automated allowing this method to scale to more games.

**Questions:**

More of a comment than a question :

There has been some work done in the chess realm to extract concepts learned by alphazero in order to teach expert humans new knowledge (https://arxiv.org/pdf/2310.16410) and there is an easy-to-use dataset + model (https://github.com/google-deepmind/searchless_chess) where value annotations are provided for board states. It might be an interesting extension of this work to see if similar shape patterns are discovered in chess networks.

---

### Note · Authors · 2024-11-22

I have read and agree with the venue's withdrawal policy on behalf of myself and my co-authors.